# Position: In Defense of Information Leakage in Concept-based Models

**Mateo Espinosa Zarlenga** [1]

## Abstract

Concept-based models (CMs), deep neural networks that ground their predictions on representations aligned with human-understandable concepts (e.g., `round`, `stripes`, etc.), have been shown to learn representations that *leak* concept-irrelevant information. As the traditional narrative goes, this leakage is undesirable and should be eradicated as it leads to uninterpretable models. In this paper, we posit that this conventional view of leakage in CMs is not only ill-posed, as the evidence of how leakage makes a model less interpretable is often inconclusive, but also bound to lead to impractical CMs under common real-world constraints. Specifically, we argue that *in real-world settings where concept incompleteness is the norm, some leakage is often necessary for constructing accurate and intervenable CMs*. To this end, we propose that there is such a thing as *benign* leakage and show that, by optimizing a reframing of the typical CM training objective, CMs can encourage and exploit this form of leakage without sacrificing accuracy or intervenability.

## 1. Introduction

The field of machine learning is teeming with phenomena whose nature has been misconstrued by early incomplete explanations. Batch Normalization (Ioffe & Szegedy, 2015) is still frequently explained as improving optimization by mitigating internal covariate shift, despite substantial empirical and theoretical evidence that this mechanism is, at best, secondary (Santurkar et al., 2018; Bjorck et al., 2018; Lipton & Steinhardt, 2019). Likewise, generalization phenomena in Deep Neural Networks (DNNs), such as grokking (Power et al., 2022), double descent (Belkin et al., 2019), and benign overfitting (Bartlett et al., 2020), are often framed as phenomena unique to DNNs, even though closely related

behaviors have been demonstrated across a range of model classes (e.g., see Wilson 2025). A similar pattern appears in common interpretations of attention: attention weights are frequently treated as causal explanations for model predictions, despite strong evidence that they do not, in general, faithfully reflect a model's decision process (Jain & Wallace, 2019; Serrano & Smith, 2019; Wiegreffe & Pinter, 2019). This paper aims to prevent the mischaracterization of another phenomenon that risks a similar trajectory, namely the effect of *information leakage* in concept-based models.

Concept-based models (CMs) (Alvarez Melis & Jaakkola, 2018; Koh et al., 2020; Yuksekgonul et al., 2023; Oikarinen et al., 2023), a rapidly growing class of methods in explainable artificial intelligence (XAI) (Poeta et al., 2023), are models that ground their downstream predictions in representations aligned with human-understandable concepts. A growing body of work has shown that the concept representations learned by CMs are often prone to *information leakage*, in which concept representations encode information that is not strictly attributable to their intended semantics (Margeloiu et al., 2021; Mahinpei et al., 2021). The prevailing narrative treats such leakage as inherently undesirable and calls for its elimination, typically arguing that leakage undermines the interpretability of CMs (Marconato et al., 2022; Havasi et al., 2022; Espinosa Zarlenga et al., 2023a; Parisini et al., 2025). In this paper, we argue that this strict view of leakage is ill-posed and counterproductive.

Importantly, we do not claim that leakage is never harmful. Rather, we argue that treating leakage as a purely negative and uncontrollable property obscures important distinctions and leads to design choices that render CMs impractical under common real-world constraints. In particular, in settings where *concept incompleteness* is the norm, enforcing strictly non-leaky representations can preclude both high task accuracy and effective test-time interventions. As there is significant evidence that both of these properties are attainable even in the presence of leakage, we take the position that **not all forms of leakage are malign, and that a controlled form of *benign* leakage can be necessary and desirable for constructing accurate and intervenable CMs**.

To substantiate this position, we first describe a general framework for analyzing CMs (Sec. 2) and use it to formalize information leakage (Sec. 3). Then, we summarize the

[1]University of Oxford, UK. Correspondence to: Mateo Espinosa Zarlenga <mateo.espinosazarlenga@trinity.ox.ac.uk>.

*Proceedings of the 43rd International Conference on Machine Learning*, Seoul, South Korea. PMLR 306, 2026. Copyright 2026 by the author(s).

dominant arguments against leakage (Sec. 4) and make the case that some degree of leakage, which we define as *benign leakage*, is often desirable if CMs are to remain useful in incomplete real-world settings (Sec. 5). Finally, we show that this form of benign leakage can be compatible with the core desiderata of CMs, including task fidelity and intervenability, provided that CMs are trained to minimize their task loss when all concepts are intervened (Sec. 6). In doing so, we show that, contrary to common narratives, leakage does not imply that a CM loses the characteristics typically associated with its interpretability and may, instead, be necessary for building accurate CMs in real-world conditions.

## 2. Background

A concept-based model (CM) $\eta : \mathcal{X} \to \mathcal{Y} \times \mathcal{C}$ is a DNN that maps features $\mathbf{x} \in \mathcal{X} \subseteq \mathbb{R}^n$ (e.g., pixels) to a *downstream task prediction* $\hat{\mathbf{y}} \in \mathcal{Y}$ (e.g., bird species) and a *concept-based explanation* $\hat{\mathbf{p}} \in \mathcal{C}$ (e.g., black_wings, medium_sized). Here, without loss of generality, we assume $\hat{\mathbf{y}} \in [0,1]^L$ is a distribution over $L$ labels and $\hat{\mathbf{p}} \in [0,1]^k$ is a vector of concept scores, with $\hat{\mathbf{p}}_i$ approximating the probability that concept $C_i$ is present in $\mathbf{x}$. Traditionally, most CMs assume access to a training dataset $\mathcal{D} = \{(\mathbf{x}^{(j)}, \mathbf{c}^{(j)}, y^{(j)})\}_{j=1}^N$, where each input is annotated with a downstream label $y^{(j)} \in \{1, \ldots, L\}$ and a vector of $k$ binary concept annotations $\mathbf{c}^{(j)} \in \{0,1\}^k$. To obtain this dataset, concept annotations may be provided by experts (e.g., radiologists' notes) or obtained via *label-free* pipelines (Oikarinen et al., 2023; Yang et al., 2023; Rao et al., 2024). Considering this setup, we now describe CBMs, a unifying framework for studying CMs.

**Concept Bottleneck Models** Given a concept-annotated dataset $\mathcal{D}$, most CMs can be framed as a Concept Bottleneck Model (CBM) (Koh et al., 2020). In its general form, seen in Figure 1, a CBM is a pair of functions $M = (g, f)$, both parameterized as DNNs, supported by a *scoring function* $s : \mathbb{R}^{k \times m} \to [0,1]^k$. The first function $g : \mathbb{R}^n \to \mathbb{R}^{k \times m}$, called the *concept encoder*, maps input features $\mathbf{x}$ to $k$ concept representations $\hat{\mathbf{c}} = (\hat{\mathbf{c}}_1, \ldots, \hat{\mathbf{c}}_k)$, where $\hat{\mathbf{c}}_i \in \mathbb{R}^m$. CBMs learn $\hat{\mathbf{c}}$ such that the $i$-th output of the scoring function $s(\hat{\mathbf{c}}) = \hat{\mathbf{p}} \in [0,1]^k$, namely $s_i(\hat{\mathbf{c}}) := \hat{\mathbf{p}}_i$, approximates $\mathbb{P}(C_i = 1 \mid X = \mathbf{x})$. Then, the *label predictor* $f : \mathbb{R}^{k \times m} \to [0,1]^L$ takes the representations $\hat{\mathbf{c}}$ and predicts a downstream task label distribution $\hat{\mathbf{y}} = f(\hat{\mathbf{c}})$.

Together, the composition $\eta = (f \circ g)$ forms a predictor $\mathbf{x} \mapsto \hat{\mathbf{y}}$ whose output can be explained by the *concept bottleneck*'s scores $\hat{\mathbf{p}} = s(g(\mathbf{x}))$. Therefore, when trained to align $\hat{\mathbf{p}}$ with $\mathbf{c}$, $\eta$ is considered interpretable. In practice, this is achieved by learning the parameters of $g$ and $f$ such that a combination of a task loss $\mathcal{L}_y(f(g(\mathbf{x})), y)$ (e.g., cross-entropy) and a concept loss $\mathcal{L}_c(s(g(\mathbf{x})), \mathbf{c})$ (e.g., binary cross-entropy) (Koh et al., 2020) is minimized. These losses

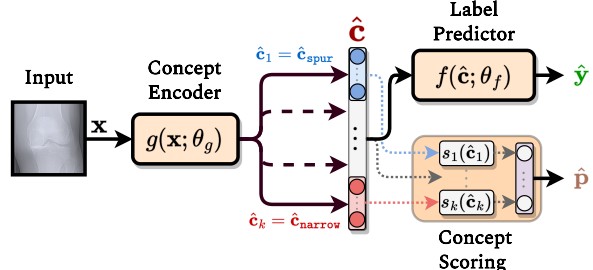

*Figure 1.* A generalized Concept Bottleneck Model (CBM).

may be optimized *independently*, *sequentially* (training $g$ before $f$), or *jointly* (minimizing a weighted sum).

The simplest instantiation of this framework is a *Sigmoidal CBM* (Koh et al., 2020), where each concept is represented by a single sigmoidal scalar ($m = 1$) and $s$ is the identity. More recent works have explored richer representations and scoring mechanisms, including embedding-based concept representations (Espinosa Zarlenga et al., 2022; Kim et al., 2023; Xu et al., 2024; Espinosa Zarlenga et al., 2025), logit-based (Koh et al., 2020) or unbounded scores (Yuksekgonul et al., 2023; Oikarinen et al., 2023), and bottlenecks augmented with residual or unsupervised side-channels (Mahinpei et al., 2021; Havasi et al., 2022; Hu et al., 2025).

**Concept Interventions** CMs that can be framed as CBMs support test-time feedback via *concept interventions* (Koh et al., 2020). An intervention on concept $i$, denoted as $g(\mathbf{x} \mid C_i := v)$, consists of modifying the concept representation $\hat{\mathbf{c}}_i$ such that it corresponds to a desired concept value $v \in [0,1]$, typically by setting $\hat{\mathbf{c}}_i := s_i^{-1}(v)$ when $s$ is invertible. The modified representation is then propagated to the label predictor, which may in turn update its task prediction.

Concept interventions are commonly framed as a mechanism for *correcting mispredicted concepts*: when predicting a high-level concept is easier than predicting the downstream task (e.g., black_wings versus bird species), a user can correct erroneous concept predictions, thereby improving task performance. However, this view understates their broader utility. Concept interventions also provide a natural interface for supplying *high-level hints* to the model, expressed in a vocabulary *shared* by experts and the model itself (i.e., the concepts). For example, through an intervention, a radiologist may indicate the presence of "bone spurs" on an X-ray before the CM processes the input features, effectively conditioning the CM's inference on prior knowledge that cannot be easily communicated through low-level feature manipulations (e.g., via pixel-level manipulations). More recent works have explored when and where interventions should be made (Shin et al., 2023; Chauhan et al., 2023; Pugnana et al., 2025), as well as how to increase the effectiveness of interventions (Espinosa Zarlenga et al.,

2023b; Steinmann et al., 2024; Vandenhirtz et al., 2024).

**Desiderata of CMs** CMs are typically designed to achieve predictive accuracy and interpretability. Although there is a lack of a task-agnostic, measurable definition of "interpretability" (Doshi-Velez & Kim, 2017), the existing literature on CMs commonly evaluates these two properties based on three measurable proxy metrics:

1. *Task Fidelity*: how accurate are the CM's downstream task predictions? (i.e., is $f(g(\mathbf{x})) \approx y$?)
2. *Concept Fidelity*: how accurate are the CM's explanations? (i.e., is $s_i(g(\mathbf{x})) \approx \mathbf{c}_i$ for all $i \in \{1, \cdots, k\}$?)
3. *Intervenability* (Laguna et al., 2024): when provided with ground-truth concepts $\mathbf{c}$, is the CM's task prediction the same as an expert would provide for $\mathbf{c}$? (i.e., does task fidelity increase as we intervene on more concepts?)

While task and concept fidelity are expected desiderata, intervenability plays a distinct role in CMs: it provides an operational test of whether the model correctly uses concepts when making predictions. By examining how predictions change under concept interventions, one can assess whether the model's reasoning aligns with that of the experts who annotated a test set with concepts. Therefore, intervenability allows us to discriminate between a CM that treats concept predictions as outputs disconnected from task inference and a model that reasons about its downstream task prediction based on the high-level concepts it has at its disposal.

Having established a framework and a set of desiderata for studying CMs, we now define *information leakage* in CMs.

## 3. Defining Information Leakage in CMs

Recent works have shown that a CM's concept representations $\hat{\mathbf{c}} = g(\mathbf{x})$ may encode information that is not strictly attributable to their intended semantics (Margeloiu et al., 2021; Mahinpei et al., 2021). For instance, by analyzing saliency maps of concept encoders in CBMs, Margeloiu et al. (2021) showed that high concept accuracy does not imply that a concept predictor relies exclusively on concept-specific features. Follow-up work by Mahinpei et al. (2021) demonstrated that representing concepts via *soft* (continuous) representations (e.g., logits or probabilities) can allow a label predictor to exploit information about other concepts or the downstream task. Related work further showed that such representations may fail to respect the spatial *locality* of concepts in the input (Raman et al., 2025). Collectively, these observations motivate the view that accurate concept prediction alone does not guarantee that concept representations form an effective bottleneck (Almudévar et al., 2025).

**Forms of Leakage** Previous works (Parisini et al., 2025; Makonnen et al., 2025) distinguish between two forms of leakage (represented in Figure 2). *Inter-concept leakage* refers to the case where the representation $\hat{\mathbf{c}}_i$ of concept $c_i$ contains information about concept $c_j$ beyond what is present in the ground-truth concept labels. Formally, this implies that $I(\hat{C}_i; C_j) > I(C_i; C_j)$, where $\hat{C}_i$ and $C_i$ denote the random variables associated with the representation and ground-truth label of concept $c_i$, respectively, and $I(\cdot; \cdot)$ is the mutual information. In contrast, *task leakage* refers to a concept representation encoding information on $Y$ that is not attributable to the concept itself, i.e., when $I(\hat{C}_i; Y) > I(C_i; Y)$, with $Y$ denoting the downstream task label. Notice that, in practice, $\hat{C}_i$ is typically continuous and high-dimensional. Therefore, the quantities characterizing both forms of leakage can only be estimated (Makonnen et al., 2025; Parisini et al., 2025) or assessed via proxies (Mahinpei et al., 2021).

**Leakage via Bypasses and Side Channels** Leakage may also arise from modeling choices in the label predictor. In particular, several CMs allow the predictor $f(\cdot)$ to exploit information outside the concept bottleneck $\hat{\mathbf{c}}$ via bypasses or residual connections of the form $f(\hat{\mathbf{c}}, \psi(\mathbf{x}))$, where $\psi(\mathbf{x})$ is an unconstrained representation of the input (Sawada & Nakamura, 2022; Yuksekgonul et al., 2023; Havasi et al., 2022). For example, Hybrid CBMs (Mahinpei et al., 2021) explicitly add $k'$ unsupervised activations $\psi(\mathbf{x}) \in \mathbb{R}^{k'}$ to the bottleneck $\hat{\mathbf{c}}$, which $f$ can exploit to make its label prediction. Such designs are typically motivated as pragmatic responses to *incompleteness* in the concept annotation set $\mathbf{c}$. In those instances, restricting the label predictor to operate on the bottleneck alone can substantially degrade task fidelity (Espinosa Zarlenga et al., 2022). From the perspective of leakage, such bypasses constitute a direct route for task-relevant information to influence predictions. Accordingly, we treat $\psi(\mathbf{x})$ as a *shared concept representation* that may carry both inter-concept and task leakage.

Now that we have formalized what leakage is, we consider how leakage has been framed in the existing literature.

## 4. Alternative Views

The CM literature holds the prevailing view that leakage is a purely negative property of CMs that should be avoided whenever possible. This has motivated methods for preventing leakage (Marconato et al., 2022), mitigating its presence (Havasi et al., 2022; Sheth & Ebrahimi Kahou, 2024; Sun et al., 2024; Ragkousis & Parbhoo, 2024; Almudévar et al., 2025), and quantifying it (Espinosa Zarlenga et al., 2023a; Marconato et al., 2024; Parisini et al., 2025; Makonnen et al., 2025; Aysel et al., 2025). Before making our case *for* leakage, we summarize the arguments commonly put forth against leakage by splitting them into three types of claims: intervenability, interpretability, and safety claims.

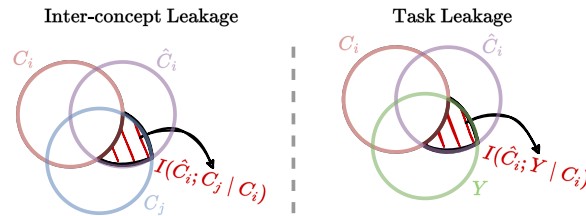

*Figure 2.* Leakage happens when the concept representation $\hat{C}_i$ encodes information that is not attributable to the ground-truth concept $C_i$. This additional information can be attributed to **(left)** another concept $C_j$ or **(right)** the downstream task $Y$.

**Intervenability Claims** A first line of argument holds that leakage undermines the *intervenability* of CMs (Havasi et al., 2022; Espinosa Zarlenga et al., 2023a; Vandenhirtz et al., 2024; Ragkousis & Parbhoo, 2024; Sun et al., 2024; Almudévar et al., 2025). These claims argue that, when concept representations encode additional information beyond their intended semantics, interventions may have unpredictable effects on downstream predictions as they may inadvertently modify that latent information (Havasi et al., 2022; Espinosa Zarlenga et al., 2023a; Parisini et al., 2025). From this perspective, leakage seems to threaten the expectation that providing correct concept labels to a CM at test time should reliably improve its task accuracy.

**Interpretability Claims** A second class of arguments asserts that leakage renders CMs fundamentally "uninterpretable" (Margeloiu et al., 2021; Mahinpei et al., 2021; Marconato et al., 2022; Ragkousis & Parbhoo, 2024; Sinha & Zhang, 2025; Almudévar et al., 2025; Makonnen et al., 2025; Sun et al., 2024; Parisini et al., 2025). These claims argue that, if a concept representation encodes information beyond its corresponding ground-truth concept, the label predictor can no longer be said to reason *only* in terms of known concepts. Nevertheless, given the lack of an agreed, measurable definition of interpretability in the context where these claims are made, these statements are often *ill-posed*. This is because it is impossible to quantify or falsify any interpretability claim without first defining under what view of interpretability we should study leakage. Hence, when evaluating these claims later, we instead measure interpretability using commonly used proxy metrics for CMs (e.g., concept fidelity, intervenability, and label-predictor weights).

A related interpretability concern is that leakage may erode the *inference value* of CMs: if the label predictor relies on information outside the concept-aligned components of $\hat{\mathbf{c}}$, a user inspecting $\hat{\mathbf{p}}$ can no longer fully reconstruct the model's reasoning (Havasi et al., 2022; Almudévar et al., 2025; Parisini et al., 2025). However, as we point out in Sections 5 and 6.2, this is an unavoidable practical limitation of any CM, leaky or non-leaky, whenever the training concept set is *incomplete* (in that case, even non-leaky CMs cannot

fully explain their predictions). As such, these arguments relate to a general issue with using CMs for inference in incomplete settings, rather than one unique to leaky models.

**Safety Claims** Finally, a small number of works argue that leakage poses *safety* risks (Marconato et al., 2022; 2024; Parisini et al., 2025). Specifically, leakage may severely affect intervenability when distribution shifts occur (Espinosa Zarlenga et al., 2025) and it may also facilitate *shortcut learning* (Geirhos et al., 2020), where the label predictor learns to exploit spurious correlations encoded in the representation. Among these claims, concerns about shortcut learning are especially important, as they may lead to models that fail to generalize and violate desirable notions of fairness and privacy (Dwork et al., 2012; Pessach & Shmueli, 2022). Nevertheless, while deserving of study in CMs, shortcuts are not unique to CMs or to concept representations. Rather, they reflect general limitations of DNN-based representation learning systems. Hence, conflating leakage with shortcut learning risks obscuring the true origin of these shortcuts and disconnecting their study from the substantial body of work on robustness (Sagawa et al., 2020; Kim et al., 2019; Sohoni et al., 2020). Because of this, while we believe these concerns are legitimate, we do not frame them as a CM-specific limitation, and we will not aim to disprove their validity in this work. Instead, we focus on CM-specific limitations arising from leakage.

Before rebutting the commonly held claims against leakage presented above, we first put forth our case *for* leakage.

## 5. The Case for Leakage

It is a common oversimplification to assume that all concept-supervised datasets are equal. However, the nature of the training concept annotations is crucial for the end quality of a CM (Collins et al., 2023; Heidemann et al., 2023; Penaloza et al., 2025). This is because some concepts are (1) inherently more informative about the downstream task (e.g., `stripes` and `fur` are more helpful in describing an animal's species than `quadruped`), (2) redundant (e.g., `black` entails `dark_color`), and (3) *necessary* for accurately predicting a downstream task (e.g., `scales` is required to classify an animal as a reptile).

Of these constraints, the most limiting is perhaps the latter. When the training concepts $\mathbf{c} \sim \mathbb{P}(C)$ lack a concept crucial for predicting a downstream task (as `scales` is for predicting reptiles), a CM may fail to achieve high task fidelity. This is particularly true if the CM's label predictor is constrained to make its prediction based only on representations that are direct proxies of the concepts' probabilities, as is the case in popular sigmoidal/soft CBMs (see Figure 3).

Therefore, underlying the setup for most CMs, there is an implicit assumption that the set of training concepts $\mathbf{c}$ is a

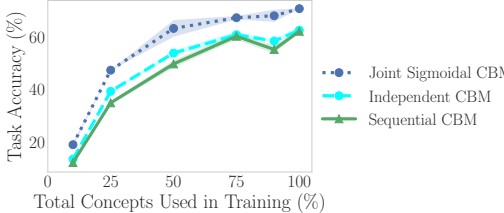

*Figure 3.* Task fidelity of sigmoidal (i.e., soft) CBMs as we vary the number $k$ of training concepts on the CUB (Wah et al., 2011) dataset (see App. B for details). CBMs that limit task leakage (e.g., independent CBMs) are more affected by incompleteness.

*complete* description (Yeh et al., 2020) of the downstream task $y$. Formally, this means CMs require that the mutual information between the downstream label $y \sim \mathbb{P}(Y)$ and the input features $\mathbf{x} \sim \mathbb{P}(X)$ given the concepts $\mathbf{c} \sim \mathbb{P}(C)$ is close to zero (i.e., $I(Y; X \mid C) \approx 0$).

In real-world concept-annotated datasets, however, *incompleteness is the typical case*. This is because it is not only costly and difficult to obtain large sets of concepts for every training sample, but it is also challenging to identify, a priori, all the important concepts one may need for future downstream tasks of interest. Moreover, although useful, label-free concept discovery pipelines are not a *sufficient* solution for incompleteness: as seen in these works' own evaluations, label-free models still tend to trail behind opaque DNNs, implying their concept sets $C$ lack information that is present in $X$ (e.g., Table 1 in Yuksekgonul et al. 2023, Table 2 in Oikarinen et al. 2023, and Table 1 in Rao et al. 2024). Furthermore, these approaches depend on the availability of (1) concepts that can be specified in natural language, and (2) foundation models that can reason about even highly domain-specific concepts and modalities, requirements which are not applicable for specialized tasks.

Taking the weak assumption that sacrificing task fidelity is highly undesirable even if this means higher interpretability, something observed across user studies (Papenmeier et al., 2019; Nussberger et al., 2022), we are then confronted by the following challenge: if we want CMs to become practical for real-world tasks where incompleteness is the norm, then these models must have a mechanism for the label predictor $f(\hat{\mathbf{c}})$ to access task-relevant information in the input features $\mathbf{x}$ that can never be present in a "pure" representation of the ground-truth concepts $\mathbf{c}$. In other words, we can only attain useful CMs in real-world incomplete settings if we encourage task leakage to capture $I(Y; X \mid C)$.

Should we therefore enable leakage in our concept representations and use it to our advantage? Or should we declare CMs potentially unsuitable for most realistic settings, where incompleteness is the norm? Here, we argue that we should enable leakage, as without it, we cannot construct CMs that can perform the one thing, for better or for worse, expected

of statistical models: to be accurate on their downstream task. Concretely, however, we argue that we should encourage a specific kind of leakage that, when properly enabled, can be exploited by a CM without compromising its expected desiderata. We call this form *benign leakage*.

### 5.1. Benign Leakage

Informally, representations $\hat{C}$ exhibit *benign leakage* if (1) they preserve enough information in them to cover for $I(Y; X \mid C)$ when $C$ is incomplete, and (2) they can be decomposed into a *concept-specific* component $\bar{C}_i$, which is the component one modifies when an intervention is performed, and *residual* component $R_i$ orthogonal to $\bar{C}_i$.

**Definition 5.1** (Benign leakage). Let $\hat{C} = (\hat{C}_1, \ldots, \hat{C}_k)$ be the concept representations produced by a CM, and assume that for each $i$ there exists a decomposition $\hat{C}_i \equiv (\bar{C}_i, R_i)$, where $\bar{C}_i$ is the concept-aligned component and $R_i$ is a residual component. Let $R := (R_1, \ldots, R_k)$. We say that $\hat{C}$ exhibits *benign leakage* if the following conditions hold:

i. **Sufficiency:** $I(Y; R \mid C) \approx I(Y; X \mid C)$

ii. **Localization:** $I\left(Y; \bar{C}_i \mid R_i, \hat{C}_{-i}\right) \approx I(Y; C_i \mid C_{-i})$

where $C_{-i} = (C_1, \cdots, C_{i-1}, C_{i+1}, \cdots, C_k)$.

Intuitively, sufficiency ensures that all label-relevant information not captured by $C$ is preserved in $\hat{C}$. In contrast, localization ensures that the information about a concept $C_i$ that is informative for predicting $Y$ can be *localized* within the $\bar{C}_i$ component of $\hat{C}_i$. Therefore, when both conditions are satisfied, a label predictor $f(\hat{\mathbf{c}})$ trained on representations with benign leakage can achieve high task fidelity in incompleteness, by exploiting information in both $R$ and $\bar{C} = (\bar{C}_1, \cdots, \bar{C}_k)$, and be intervenable, by ensuring it uses the well-localized variables $\bar{C}$ to access information in $C$.

We point out that without considering how $f$ operates or learns, we cannot make claims about a CBM's intervenability based only on properties of $\hat{C}$. For example, a Hybrid CBM may very well exhibit benign leakage, as it cleanly separates the activations that encode concept predictions from those that carry additional leakage. Yet, there is strong evidence that these models, trained only on a joint CBM loss, are not intervenable (Espinosa Zarlenga et al., 2022). Because of this, benign leakage, or, for that matter, any property of the concept predictions alone, cannot alone imply that a CM is intervenable, as intervenability is also a property of the label predictor. Nevertheless, what benign leakage allows us to ascertain is that well-conditioned label predictors *can* be accurate and intervenable (i.e., it is a *necessary* condition for task fidelity and intervenability).

**Optimizing for Benign Leakage**    Given our definition of benign leakage, we notice that, under standard well-specification assumptions (see App. A for proof), achieving

sufficiency for a CBM $(g, f)$ is equivalent to minimizing the negative task likelihood under full concept interventions:

$$\mathcal{L}_{\text{int}} = \mathbb{E}_{(\mathbf{x}, \mathbf{c}, y) \sim \mathcal{D}} \big[ - \log \mathbb{P}_f \big( y \mid \hat{C} = g(\mathbf{x} \mid C := \mathbf{c}) \big) \big]$$

$$= \mathbb{E}_{(\mathbf{x}, \mathbf{c}, y) \sim \mathcal{D}} \Big[ \mathcal{L}_y \Big( f \big( g(\mathbf{x} \mid C := \mathbf{c}) \big), y \Big) \Big]$$

Intuitively, $\mathcal{L}_{\text{int}}$ can be seen as a generalization of independent CBM training. As such, $\mathcal{L}_{\text{int}}$ is similar to the "prior loss" used in (Espinosa Zarlenga et al., 2025) to learn robust concept representations. The difference lies in the fact that $\mathcal{L}_{\text{int}}$ applies to any CM, not just those that decompose their representations into exogenous and endogenous variables. This makes $\mathcal{L}_{\text{int}}$, which only requires an additional forward pass of $f(\cdot)$, an *efficient* way to ensure sufficiency[1].

In contrast, ensuring localization is not as easy as ensuring sufficiency. When one can explicitly decompose a CM's concept representation $\hat{C}_i$ into a concept-aligned component $\bar{C}_i$ and a residual "leaky" component $R_i$ (e.g., as in Hybrid CBMs (Mahinpei et al., 2021), Residual PCBMs (Yuksekgonul et al., 2023), and MixCEMs (Espinosa Zarlenga et al., 2025)), localization may be directly optimized if the mutual information terms are quantities that can be estimated (a likely intractable problem if $\hat{C}$ is high-dimensional and not well-constrained). If a CM does not explicitly decompose its concept representations, then optimizing directly for localization becomes even more difficult with an architecture-agnostic regularizer such as $\mathcal{L}_{\text{int}}$. Nevertheless, we can still introduce implicit incentives that encourage localization. As we argue in the next section, due to the tendency of DNNs to learn simpler hypotheses, such an incentive may be a by-product of minimizing the sufficiency regularizer $\mathcal{L}_{\text{int}}$.

# 6. Revisiting the Arguments Against Leakage

Having argued that benign leakage is necessary for building accurate and intervenable CMs that can operate in incompleteness, we now make the case that information leakage does not *necessarily* affect the CM's intervenability or interpretability (under traditional proxy metrics).

## 6.1. Revisiting Intervenability Claims

The first evidence against the common claim that information leakage leads to unintervenable CMs comes from previous works themselves: as recreated in Figure 4 (left), several CMs designed to enable leakage, either by using dynamic embedding representations, such as Concept Embedding Models (CEMs) (Espinosa Zarlenga et al., 2022) and Probabilistic CBMs (ProbCBMs) (Kim et al., 2023), or by using *small* residual bypasses, such as Hybrid CBMs (Mahinpei et al., 2021), are more or similarly intervenable to

CBMs commonly agreed to have very minimal leakage (i.e., independently-trained sigmoidal CBMs). Similar results have been observed for other leakage-enabling CMs such as residual Autoregressive CBMs (Havasi et al., 2022) (and related information-theoretic ways for selecting maximally informative concept subsets (Chattopadhyay et al., 2022)), Semi-supervised CBMs (Hu et al., 2025), and further variants of CEMs (Espinosa Zarlenga et al., 2023b; 2025). Therefore, the common claim that leakage *necessarily* precludes a CM's intervenability is false.

It is important to notice, however, that it is true leakage *can* affect intervenability: as seen in Figure 4 (right), some CMs that enable leakage become almost entirely unintervenable when facing incompleteness (something we do not necessarily see when they are deployed in an equivalent, but complete setting as seen in App. C). Therefore, this begs the question: *when is a leakage-enabling CM intervenable?* As discussed next, this is possible when a CM (sometimes implicitly) optimizes for sufficiency (i.e., $\mathcal{L}_{\text{int}}$).

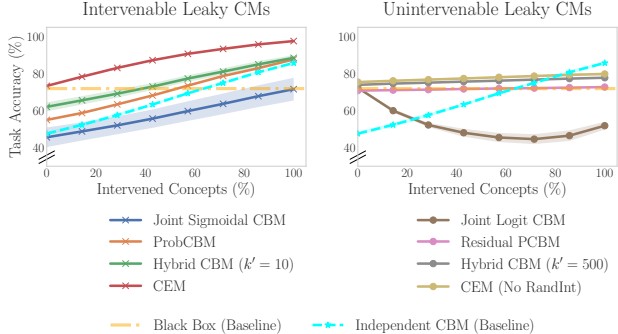

*Figure 4.* Task accuracy (y-axis) of leaky CMs trained on an *incomplete* version of CUB ($k = 22$, details in App. B) as we intervene on concepts at test time (x-axis). **(left)** Leaky CMs can remain intervenable in incompleteness. **(right)** Some leaky variants, however, lose their intervenability (curves are non-increasing).

**Simplicity Bias and Benign Leakage**  CMs that enable leakage can remain highly accurate and intervenable in incomplete settings if (1) interventions do not destroy the leakage (e.g., by not overwriting the activations that may be carrying the leaked information), *and* (2) they implicitly encourage forms of benign leakage. This is true, for example, for embedding-based approaches such as CEMs, ProbCBMs, and MixCEMs, where interventions are operations that swap one dynamic high-dimensional representation for another without destroying leakage. All of these models are randomly intervened on during training, a training-time mechanism called *RandInt* which can be seen as implicitly minimizing the *sufficiency* regularizer $\mathcal{L}_{\text{int}}$. Similar training pipelines with loss terms that more closely resemble $\mathcal{L}_{\text{int}}$ can be found in intervention-aware versions of CEMs (Espinosa Zarlenga et al., 2023b).

However, as discussed in Sec. 5.1, *sufficiency* alone does

---

[1]For example, introducing $\mathcal{L}_{\text{int}}$ when training a CEM (Espinosa Zarlenga et al., 2022), without much code optimization, increased the total training time by only 10%.

not guarantee benign leakage. For a label predictor to be intervenable when leakage is present, the variables encoding concept-specific information must be well-localized within $\hat{C}$. Hence, if these leakage-enabling CMs only (implicitly) minimize $\mathcal{L}_{\text{int}}$, what leads them to also appear to attain *localization*? Moreover, even if concepts are localized, what encourages label predictors to properly attend to changes in these variables and become intervenable? Here we hypothesize that both of these effects can be seen as a consequence of the *simplicity bias* of DNNs: when given an option to learn several competing hypotheses, stochastic gradient descent favors simpler hypotheses over more complex ones (Pérez et al., 2019; Shah et al., 2020). Specifically, we argue that localization and intervenability naturally arise when jointly minimizing $\mathcal{L}_{\text{int}}$ because concept encoders and label predictors that achieve these properties are simpler than equally-performing models that do not.

To see this, consider, without loss of generality, what happens when we train a CM to minimize the traditional joint CBM loss together with the regularizer $\mathcal{L}_y(f(g(\mathbf{x} \mid C_i := \mathbf{c}_i)), y)$ (for some fixed concept index $i$). Here, whenever we perform an intervention $g(\mathbf{x} \mid C_i := \mathbf{c}_i)$, we are changing in some way the output representation $\hat{\mathbf{c}}_i$ of concept $C_i$ (e.g., by fixing $\hat{\mathbf{c}}_i = c_i$ for sigmoidal CBMs). Therefore, when tasked to minimize the task loss given the ground-truth concept label $\mathbf{c}_i$, a label predictor that extrapolates how $\hat{\mathbf{c}}$ is affected by the intervention $g(\mathbf{x} \mid C_i := \mathbf{c}_i)$ (i.e., how $\hat{\mathbf{c}}$ encodes the true value of $\mathbf{c}_i$) will only require its concept predictor to learn to encode the information $I(Y; X \mid C_i)$ in the rest of its representations to be able to accurately predict $Y$. Learning this label predictor and concept encoder is arguably simpler than learning a concept encoder that needs to both (a) accurately predict $C_i$ and re-encode its information somewhere new in $\hat{\mathbf{c}}$, and (b) still learn to encode the information $I(Y; X \mid C_i)$ in $\hat{\mathbf{c}}$. Therefore, when $\mathcal{L}_{\text{int}}$ is heavily penalized, both the label predictor and concept encoder will gravitate towards learning the simpler hypothesis whose label predictor (1) effectively extrapolates which variables in $\hat{\mathbf{c}}$ are affected by the intervention on concept $C_i$ (i.e., leading to localization), and (2) uses this information to quickly minimize $\mathcal{L}_{\text{int}}$ (i.e., leading to intervenability).

**The Effects of Proper Conditioning** We note that indirect evidence for the hypothesis above already exists in the literature: Espinosa Zarlenga et al. (2025) show in their appendix that, when training a leaky CM using a stricter variant of $\mathcal{L}_{\text{int}}$ and a task loss term, the CM learns to implicitly align each ground-truth concept $C_i$ with the set of variables that are affected when concept $C_i$ is intervened on. This can be done even when **no explicit concept loss was used** to train the CM. We illustrate the strength of this effect in Figure 5. There, we visualize the result of intervening on an otherwise *opaque DNN* that was trained with the regularizer $\mathcal{L}_{\text{int}}$. To achieve this, during training, we selected a set of $k$ neurons

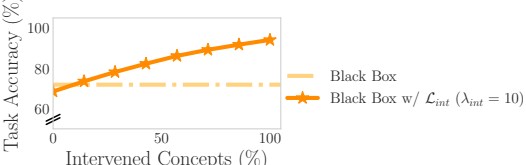

*Figure 5.* A DNN trained on an *incomplete* CUB task ($k = 22$) w/o *any* concept alignment loss becomes intervenable with $\mathcal{L}_{\text{int}}$.

in the penultimate layer of the DNN and intervened on them as if they were normal CBM soft concept representations (which they are not, as we do not introduce any concept alignment losses when training this model, and the neurons do not form a bottleneck). We see that this DNN, trained without *any* concept alignment loss, is intervenable and shows signs of localization as measured by the ROC-AUC between the neuron $h_i$ used to intervene on concept $C_i$ and the ground-truth $\mathbf{c}_i$ (we get a mean concept ROC-AUC of $80.79 \pm 0.11\%$). Hence, even though directly optimizing for *localization* is intractable, by heavily penalizing $\mathcal{L}_{\text{int}}$ we can implicitly encourage both localization and sufficiency.

Moreover, we observe that the localization resulting from explicitly minimizing $\mathcal{L}_{\text{int}}$ can be exploited to make leakage-enabling CBMs intervenable. As shown in Figure 6, by adding a strong weight to $\mathcal{L}_{\text{int}}$ when training Joint Logit CBMs, large Hybrid CBMs, and CEMs without RandInt, we can make all of these models intervenable, something we saw did not naturally occur in Figure 4 (right) (App. D shows similar results for other CMs). Perhaps the most surprising result is that even large Hybrid CBMs become highly intervenable when minimizing $\mathcal{L}_{\text{int}}$; all without sacrificing task or concept fidelity (something that *does not* happen when RandInt is introduced in their training, as seen in Espinosa Zarlenga et al. 2022). Therefore, these results strongly suggest that, when CMs are properly conditioned by penalizing $\mathcal{L}_{\text{int}}$, *information leakage does not necessarily lead to a loss of intervenability*. In fact, it may lead to intervenable CMs with higher task fidelities.

We note that we observe the same trends as in Figures 5 and 6 across several different tasks (see App. E).

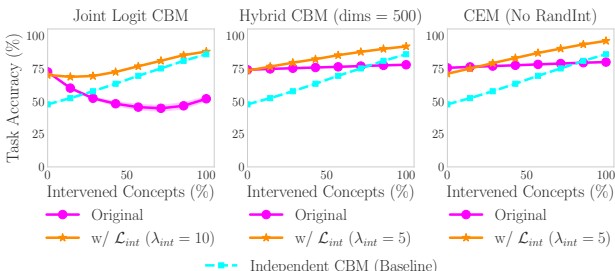

*Figure 6.* Effect of regularizing previously unintervenable leaky CMs with $\mathcal{L}_{\text{int}}$ on an *incomplete* version of CUB ($k = 22$).

## 6.2. Revisiting Interpretability Claims

We now examine the interpretability claims against leakage. For this, we use as an example the Hybrid CBM $M_H = (g_H, f_H)$ trained with $\lambda_{\text{int}} = 5$ on the incomplete CUB task of Figure 6. Given the number of unsupervised activations $\psi(\mathbf{x}) \in \mathbb{R}^{k'}$ in $M_H$'s bottleneck $\hat{\mathbf{c}}$ ($k' = 500$, much larger than the concept-aligned activations $k = 22$), this model has an expressive mechanism for leaking information in $\hat{\mathbf{c}}$. As such, we use $M_H$ to assess the validity of interpretability claims against leakage. We note that we assess these claims using the same proxy metrics that prior work uses to study interpretability (e.g., concept ROC-AUC, model weights) only to show that, from the perspective of those common metrics, well-conditioned leaky CMs do not lose properties usually associated with CMs considered "interpretable". With this in mind, we summarize our findings below.

**Traditional metrics do not indicate leaky CMs are less interpretable** Interpretability claims against leakage typically argue that if a concept representation, or the concept bottleneck $\hat{\mathbf{c}}$ more generally, encodes information beyond its corresponding concepts, then the CM is less "interpretable". However, our experiments above suggest that, under the same metrics used by previous works to evaluate interpretability, this conclusion is not obvious: the Hybrid CBM $M_H$ above has a very similar concept fidelity to the non-leaky Independent CBM $M_I$ baseline ($M_H$ has a mean concept ROC-AUC of $93.65 \pm 0.22\%$ while $M_I$'s is $93.90 \pm 0.11\%$). Moreover, as seen in Figure 6, the Hybrid CBM has (1) a significantly higher task fidelity than $M_I$ ($73.31 \pm 0.26\%$ vs $47.61 \pm 1.01\%$), and (2) a significantly higher area under the intervention curve than $M_I$ (implying higher intervenability). These curves also reveal that $M_H$ likely allows leakage at its bottleneck: its task accuracy when it receives several interventions is higher than the task accuracy achieved by the independent CBM when it is intervened on all training concepts (i.e., the rightmost point of the Independent CBM's intervention curve).

The results above suggest that, from the perspective of the evaluation metrics used to study CMs, a leaky model like $M_H$ is likely to be considered "better" than a non-leaky model like $M_I$. This, however, is not aligned with common views of leakage. It is then natural to wonder under what perspective one can argue that a leaky CM is less interpretable than a non-leaky counterpart. For example, the traditional CM evaluation metrics used above may not detect whether a label predictor operating on leaky representations reasons about a task $y$ using concepts the same way an expert would, a common argument against leakage. Therefore, could we somehow evaluate this property in a different manner?

As the label predictors for $M_H$ and $M_I$ are linear models, we can study this question by looking at how a leaky CM's task predictor weighs concepts $C$ to predict $Y$ and compare

it to how a model we consider "interpretable", such as $M_I$, performs the same task. Let the label predictor of $M_A$, for $A \in \{H, I\}$, be $f_A(\hat{\mathbf{c}}) = \text{Softmax}(W_A \hat{\mathbf{c}} + b_A)$, where $W_A \in \mathbb{R}^{L \times \dim(\text{Dom}(f_A))}$. Moreover, let $T_y(M_A)$ be the set of top-$m$ weights in the $y$-th row of $W_A$ (we use $m = 5$, but show in App. F that our results hold across $m$). By looking at the label predictors of $M_H$ and $M_I$, we see that, on average, the size of $O = |T_y(M_H) \cap T_y(M_I)|$ is $4.31 \pm 0.86$. In other words, the set of top-5 most important concepts used by $M_I$'s label predictor to predict a given class $y$ is, on average, almost the same as the set of top-5 most important bottleneck *activations* used by $M_H$ when predicting the same class. To place this within a reasonable frame of reference: the overlap of $T_y(M_I)$ for two differently-seeded Independent CBMs is $4.45 \pm 0.75$, not statistically different from $O$. In contrast, the same value when using an ill-conditioned Hybrid CBM with $\lambda_{\text{int}} = 0$ is $2.01 \pm 0.21$, which is significantly lower than $O$. Given that the set of potential activations $M_H$ has in its bottleneck is much larger than the set of ground-truth concepts ($(k + k') = 522 > 22 = k$), these results are not only surprising but also a strong indication that $M_H$'s use of concepts is similar to that of $M_I$, a non-leaky model. Therefore, if models such as $M_I$ are considered "interpretable", why should we not do the same for well-conditioned leaky models like $M_H$?

Altogether, the simple experiments above suggest that it is unclear under which definition of interpretability leakage leads to *less* interpretable models in incomplete settings. As most interpretability concerns about leakage fall in this ill-specified setup, they are therefore unfalsifiable.

**Partial explanations are useful explanations** Finally, even if leakage may lead to models that use information not in $C$ to predict $Y$, they are still *useful*: they can still at least *partially* explain their predictions based on a potentially incomplete set of concepts $C$, concepts which experts themselves also use to explain, sometimes also partially, their reasoning. Moreover, as seen in Section 6.1, leakage does not preclude intervenability, enabling experts to provide test-time concept-based feedback to leaky but well-conditioned CMs (e.g., CMs that minimize $\mathcal{L}_{\text{int}}$). In the common setting where $C$ is incomplete, the alternative intervenable model would be to use a perfectly non-leaky CM that, at best, has the same power to partially explain its outputs at the cost of a severe drop in its task fidelity. Given a choice between these two options, we struggle to see an argument for selecting the non-leaky CM over the more accurate leaky CM.

## 7. Discussion and Conclusion

### 7.1. A Unifying View of Existing Inductive Biases

$\mathcal{L}_{\text{int}}$ explicitly encapsulates inductive biases already present, in different forms, across the CM literature: Independent

CBMs train $f$ on a fully-intervened bottleneck, making it the limit of $\mathcal{L}_{\text{int}}$ as $\lambda_{\text{int}} \to \infty$; CEMs and ProbCBMs stochastically intervene on concept subsets during training, approximating $\mathcal{L}_{\text{int}}$ in expectation; architectures that predict concepts iteratively (Havasi et al., 2022; Chattopadhyay et al., 2022) implicitly penalize models when they make mistakes given all concept labels, as in $\mathcal{L}_{\text{int}}$; finally, intervention-aware losses (Espinosa Zarlenga et al., 2023b), KL regularizers (Almudévar et al., 2025), and robustness losses (Espinosa Zarlenga et al., 2025), all implicitly encourage sufficiency or localization. $\mathcal{L}_{\text{int}}$ therefore acts as an architecture-agnostic regularizer that recovers several previously proposed mechanisms as special cases.

### 7.2. Relation to Other Subfields

**Sparse Autoencoders and Steering**   Sparse autoencoders (SAEs) (Cunningham et al., 2023) are commonly used to discover human-interpretable features in an LLM's representations. A central goal of this line of work is *monosemanticity*, i.e., discovering features that respond to a single concept (Bricken et al., 2023). Our localization condition in Definition 5.1 can be seen as a formalization of monosemanticity: information about a concept should be localized to a known component of the representation.

Under this view, steering on monosemantic SAE features is a form of benign leakage that can produce targeted changes in LLM behaviors (Templeton, 2024; Arad et al., 2025). This suggests that the CM and mechanistic interpretability (MechInterp) literatures share the common goal of learning representations in which concepts are localized. The difference, however, lies in the fact that the MechInterp literature treats leakage as an unavoidable property of models and focuses on learning to localize it, rather than eliminating it, as in the CM literature. We note that similar connections exist with works for concept erasure (Ravfogel et al., 2022) and representation engineering (Zou et al., 2023; Li et al., 2023), where we are interested in modifying components of a model's internal representation such that we can control a specific concept (i.e., localize it) without damaging the model's downstream task utility (i.e., attain sufficiency).

We therefore hope that, by recognizing that leakage cannot be avoided and that one must instead focus on controlling it, future CM work may be more open to exploring methodologies in which leakage is expected and required (as in works on mechanistic interpretability). Doing so would enable CMs to be applied to practical settings where traditional steering and SAEs are currently used.

**Concept-based Foundation Models**   Label-free concept discovery pipelines (Oikarinen et al., 2023; Rao et al., 2024; Yang et al., 2023) that build CMs on top of foundation models operate in precisely the regime where our argument for benign leakage applies most strongly: concept sets extracted from foundation models are inherently incomplete, and their representations encode far more information than that captured by the discovered concept set. Embracing benign leakage rather than attempting to eliminate it is, therefore, not only pragmatically necessary in this setting but also a plausible path to scaling CMs. Hence, we believe it is important to adjust the current narrative against leakage if the field is to enable the construction of powerful, concept-based foundation models. Otherwise, continuing to treat leakage as something that must be removed at all costs risks making this class of models unappealing to the community before their potential has been fully explored.

### 7.3. Conclusion

This paper argued that the prevailing view of information leakage in CMs as a purely negative phenomenon is, in many cases, ill-posed and counterproductive. Specifically, we made our case by showing, through a set of simple experiments and evidence from prior work, that previous claims that leakage leads to CMs that (1) are not intervenable, or (2) sacrifice properties usually associated with interpretable models, are, at best, inconclusive. In contrast, we argued that there are good reasons to want some leakage: in realistic concept-incomplete settings, eradicating leakage will only preclude CMs from remaining accurate, thereby making these models impractical. Hence, we showed that minimizing the CM's task loss when all concepts are intervened can lead to leakage that is properly *localized* and *sufficient* to overcome incompleteness. Such a simple regularizer, applicable to all CMs, enables even models previously thought to be unintervenable to remain accurate and intervenable, while retaining common proxy properties usually associated with the interpretability of CMs, such as high concept fidelity and meaningful concept-to-task weight structure.

### 7.4. Call to Action

Taken together, we hope that the arguments and evidence put forth in this work naturally form a call for the CM community to (1) see past the goal of *entirely* eliminating leakage and instead explore ways to better *control* it, (2) evaluate new CMs on incomplete settings, reporting intervenability under such conditions and providing evidence for how the CM performs as the degree of incompleteness varies, (3) avoid broad claims that leakage or any other property of interpretable models *necessarily* destroys interpretability without specifying falsifiable criteria, (4) separate CM-specific questions and phenomena (e.g., leakage) from well-studied, yet more general failure modes of statistical learning (e.g., shortcut learning), and (5) develop principled, architecture-agnostic ways to measure benign leakage.

## Acknowledgments

We would like to thank Pietro Barbiero, Oscar Hill, Naveen Raman, and Andrei Margeloiu for their helpful feedback and discussions on earlier iterations of this manuscript.

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

## A. Proof of Sufficiency Optimization

Below we formalize the claim made in Section 5.1 that, under a reasonable set of assumptions, minimizing the task loss when all concepts are intervened is equivalent to achieving *sufficiency* in the learned concept representations $\hat{C}$. For this, we first notice that, due to the data processing inequality, it must be the case that, for localized concept representations $\hat{C} = (\bar{C}, R)$, we must have $I(Y; R \mid C) \leq I(Y; X \mid C)$, since $R$ is a component of $\hat{C}$, which is in turn a function of $X$. Therefore, to achieve sufficiency, which we defined as $I(Y; R \mid C) \approx I(Y; X \mid C)$, what we want is to maximize $I(Y; R \mid C)$ so that it approaches its upper bound $I(Y; X \mid C)$. Hence, under this perspective, the following theorem says that this maximization is possible by minimizing the task loss when we intervene on all concepts of the bottleneck:

**Theorem A.1** (Sufficiency Optimization). *Let* $M = \big(g(\cdot\,;\theta_g), f(\cdot\,;\theta_f)\big)$ *be a CBM and* $(X, C, Y) \sim \mathcal{P}$ *be the data-generating distribution. Let* $Z := g(X \mid C := C\,;\theta_g)$ *denote the random variable corresponding to the* fully intervened *bottleneck. Assume concept interventions are localized so that* $Z \equiv (C, R)$ *for some residual random variable* $R$. *If* $\mathcal{L}_y$ *is the cross-entropy loss and* $f$ *is well-specified for* $\mathbb{P}(Y \mid Z)$, *then:*

$$\underset{\theta_g, \theta_f}{\arg\min} \underbrace{\mathbb{E}_{(\mathbf{x},\mathbf{c},y)\sim\mathcal{P}}\left[\mathcal{L}_y\left(f\Big(g\big(\mathbf{x} \mid C := \mathbf{c}\big)\Big), y\right)\right]}_{\text{All-concepts-intervened task loss}} \equiv \underset{\theta_g}{\arg\max} \underbrace{I\big(Y; R \mid C\big)}_{\text{Sufficiency}} \tag{1}$$

*Proof.* Fix $\theta_g$ and let $q_{\theta_f}(y \mid z)$ denote the conditional distribution induced by $f(\cdot; \theta_f)$. By the standard cross-entropy decomposition, we must have that the LHS of Equation 1 becomes:

$$\mathbb{E}\Big[-\log q_{\theta_f}(Y \mid Z)\Big] = H(Y \mid Z) + \mathbb{E}\Big[D_{\mathrm{KL}}\Big(\mathbb{P}(Y \mid Z) \,\|\, q_{\theta_f}(Y \mid Z)\Big)\Big]$$

Under well-specification, there exists a set of parameters $\theta_f^\star$ such that $q_{\theta_f^\star}(Y \mid Z) = \mathbb{P}(Y \mid Z)$, making the KL divergence term in the expression above zero. Hence, for any fixed $\theta_g$ we must have that $\min_{\theta_f} \mathbb{E}\big[\mathcal{L}_y(f(Z;\theta_f), Y)\big] = H(Y \mid Z)$.

This implies that the LHS of Equation 1 can be equivalently written as follows:

$$\begin{aligned}
\underset{\theta_g, \theta_f}{\arg\min} \mathbb{E}\big[\mathcal{L}_y\big(f(Z;\theta_f), Y\big)\big] &\equiv \underset{\theta_g}{\arg\min}\ H(Y \mid Z) \\
&= \underset{\theta_g}{\arg\min} H(Y \mid C, R) && \text{(By our localization assumption)} \\
&= \underset{\theta_g}{\arg\min} \Big(H(Y \mid C) - I(Y; R \mid C)\Big) \\
&\equiv \underset{\theta_g}{\arg\min} -I(Y; R \mid C) && \Big(\text{As } H(Y|C) \text{ does not depend on } \theta_g\Big) \\
&\equiv \underset{\theta_g}{\arg\max} I(Y; R \mid C)
\end{aligned}$$

This directly yields the equivalency we wished to show in Equation 1. $\qquad\square$

## B. Experimental Details

In Sections 5 and 6, we made use of some simple experiments to build our case. In this appendix, we discuss details for recreating these results.

### B.1. Datasets and Tasks

**General Setup**  In all of our illustrative experiments in Sections 5 and 6, we use the CUB-200 dataset (Wah et al., 2011), preprocessed by Koh et al. (2020). Each sample $(\mathbf{x}^{(j)}, \mathbf{c}^{(j)}, y^{(j)})$ in this original dataset corresponds to a normalized RGB image $\mathbf{x}^{(j)} \in [0, 1]^{3 \times 299 \times 299}$ of a bird that is annotated with one of $L = 200$ bird species $y^{(j)}$. Each sample comes with a set of 112 binary concept annotations $\mathbf{c}^{(j)} \in \{0, 1\}^{112}$ representing visual attributes of the bird (e.g., black_wing, medium_sized, etc.). By construction, the concept set in this dataset provides a complete description of the downstream label $y$ (as each task label in the CUB formulation by Koh et al. (2020) is assigned its own "concept profile"). These concepts are grouped into 28 mutually exclusive concept groups (e.g., wing_color, size). Therefore, when intervening on models trained on this dataset, we apply the intervention to all concepts within the same group simultaneously. Finally, we use the train-validation-test splits from (Koh et al., 2020) and randomly flip and crop images during training.

**Used Tasks** To create an incomplete training set, following Espinosa Zarlenga et al. (2023b), we subsample the training concept annotations of the CUB dataset described above to construct an incomplete task we call *CUB-Incomplete*. Specifically, we subsample, at random, 25% of the 28 original concept annotation groups, yielding an *incomplete version of CUB* whose concept vectors **c** are formed by 7 concept groups that have, together, a total of $k = 22$ concepts.

Unless otherwise stated or clear from the context (e.g., the results of Figure 3), all experiments are conducted on CUB-Incomplete. The only exceptions are the experiments in App. C, which use the concept-complete version of CUB to explore how incompleteness affects the intervenability of leakage-enabling models, and those in App. E, where we show that our observations on CUB generalize to other tasks.

### B.2. Baselines

In our illustrative experiments, we train and evaluate the following baselines:

1. **Concept Bottleneck Models** (Koh et al., 2020): We train CBMs with sigmoidal (i.e., $\hat{\mathbf{c}}_i \in [0, 1]$ and $s(\hat{\mathbf{c}}) = \hat{\mathbf{c}}$) and logit (i.e., $\hat{\mathbf{c}}_i \in \mathbb{R}$ and $s(\hat{\mathbf{c}}) = \sigma(\hat{\mathbf{c}})$) bottlenecks. These models are trained *independently*, *sequentially*, and *jointly* (minimizing the combination of $\mathcal{L}_y(\cdot) + \lambda_c \mathcal{L}_c(\cdot)$). In our experiments, we use the independently trained sigmoidal CBM (*Independent CBM*) as a minimal-leakage baseline, since its label predictor is trained on ground-truth concepts.

2. **Hybrid CBMs** (Mahinpei et al., 2021): We include Hybrid CBMs as an example of a leaky CM that introduces a bypass channel in its bottleneck. In practice, we train these models by extending a sigmoidal CBM with $k'$ additional unconstrained activations $\psi(\mathbf{x})$ which are added to the bottleneck. This model is trained by minimizing a joint loss that weights the concept loss term using a hyperparameter $\lambda_c$.

3. **Black Box (Baseline)**: We implemented a Black Box DNN baseline using a Hybrid CBM with $k' = 500$ additional neurons in its bottleneck whose concept loss weight during training is set to 0 (i.e., $\lambda_c = 0$). This ensures the Black Box model is given the same capacity as competing models it is compared against (e.g., equivalent Hybrid CBMs).

4. **Concept Embedding Models (CEMs)** (Espinosa Zarlenga et al., 2022): We include, as an example of a highly leaky baseline, a CEM that represents concepts as a mixture of two dynamic embeddings with dimensionality $m = 16$. Unless explicitly stated otherwise, all CEMs are trained using RandInt, as in their original work. That is, during training, we intervene on a concept with probability $p_{\text{int}} = 0.25$.

5. **Probabilistic CBMs (ProbCBMs)** (Kim et al., 2023): As a second embedding-based baseline, we use ProbCBMs, which represent concepts with probabilistic embeddings of size $m = 16$. As suggested by the authors, we train ProbCBMs with RandInt, intervening on a concept during training with probability $p_{\text{int}} = 0.5$.

6. **Post-hoc CBMs (PCBMs)** (Yuksekgonul et al., 2023): Finally, we include as baselines PCBMs trained from the Black Box model baseline. We train both a standard Post-hoc CBM, whose leakage is relatively small, and a Residual Post-hoc CBM, which introduces an explicit residual channel that enables higher task accuracy in incompleteness at the cost of higher leakage.

**Model Selection and Implementation** All models are implemented in PyTorch (Paszke et al., 2019) and trained using the same backbone architectures, optimizers, and training schedules as in Espinosa Zarlenga et al. (2025). Specifically, we made use of and extended the publicly available code[2] from Espinosa Zarlenga et al. (2025) to access implementations for all the baselines used in our experiments. To better communicate our position, we avoid expensive fine-tuning of baselines by reusing the exact hyperparameters and model selection reported in (Espinosa Zarlenga et al., 2025) for each baseline. Hence, unless specified otherwise, all baselines were built from an ImageNet-pretrained ResNet-18 (He et al., 2016) backbone for the concept encoder $g(\cdot)$, and a linear layer for the label predictor $f(\cdot)$. For further details on the hyperparameters used for each baseline in our CUB experiments, we refer the reader to Appendix D of (Espinosa Zarlenga et al., 2025).

**Sufficiency Regularizer** When introducing the sufficiency regularizer $\mathcal{L}_{\text{int}}$, we add it to the training objective with weight $\lambda_{\text{int}} \in \{0, 1, 5, 10\}$ while keeping all other hyperparameters and training configurations of the underlying model constant. The value of $\lambda_{\text{int}}$ is then selected based on the area under the intervention curve on the validation set. For the incomplete CUB experiments, this yields $\lambda_{\text{int}} = 10$ for most models, except for Hybrid CBMs and CEMs, where it selects $\lambda_{\text{int}} = 5$.

---

[2]See https://github.com/mateoespinosa/cem for configuration files to recreate our results.

When we train the Black Box baseline model with the regularizer $\mathcal{L}_{\text{int}}$, we select $k$ neurons in the output of the penultimate layer and intervene on those neurons as we would for a traditional sigmoidal CBM (i.e., setting the output of the neuron to $\hat{c}_i := c_i$). Furthermore, when using this regularizer for models whose concept representations are unbounded scalars (e.g., logit CBMs (Koh et al., 2020)), we intervene on concepts by setting their respective representations to a high logit value when $c_i = 1$ (e.g., we use $+5$) and to a low logit value when $c_i = 0$ (e.g., we use $-5$).

## C. Completeness Experiments

To disentangle the effects of information leakage from those of concept incompleteness, we replicate the intervention experiments of Figure 4 on a complete version of CUB (using the same setup as Koh et al. (2020) where $k = 112$).

Our results, summarized in Figure 7, suggest that, when the concept set is complete, all leakage-enabling models remain intervenable, including those that were seen to lose intervenability when the concept set was incomplete (Figure 4). Nevertheless, we observe that even in this instance, the degree of intervenability of the leaky CMs in the right panel of Figure 7 is not the same: for example, although CEMs without RandInt do increase their task accuracies as they are intervened on, the effect is minimal compared to that of other models. This implies that it is likely that incompleteness can lead to several leaky models to become unintervenable, but its effects on leaky CMs can vary.

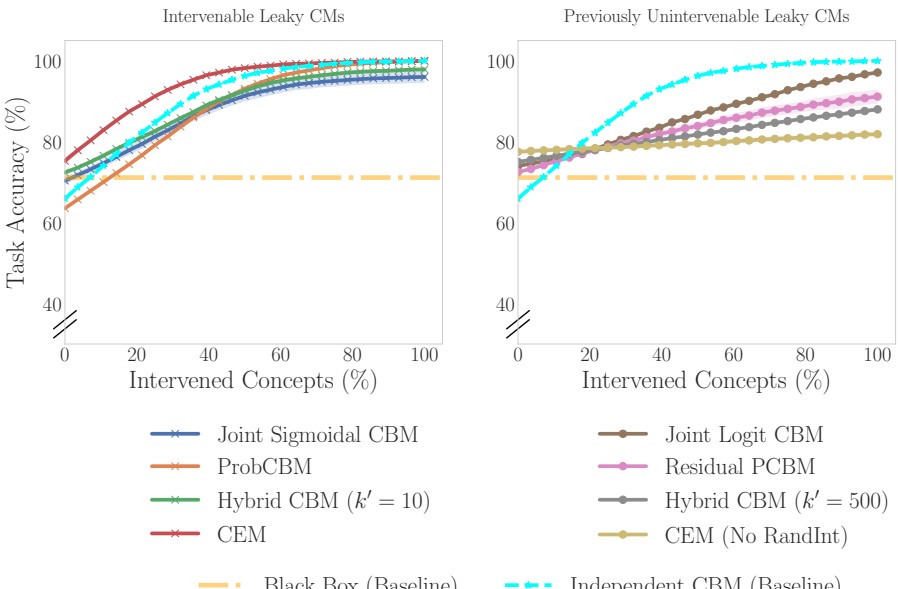

*Figure 7.* Intervention curves for leakage-enabling CMs on a *complete* version of CUB ($k = 112$, as in Koh et al. 2020). **(left)** Leakage-enabling CMs that were intervenable in the incomplete version of CUB remain highly intervenable here. **(right)** Leakage-enabling CMs that were previously unintervenable in the incomplete version of CUB (Figure 4) can become intervenable when the concept set is complete, although to a lesser degree than the non-leaky baseline (Independent CBMs) variants.

## D. Extended Experiments With Sufficiency Regularizer

We further evaluate the effect of the sufficiency regularizer $\mathcal{L}_{\text{int}}$ on models that were previously unintervenable. Specifically, we apply $\mathcal{L}_{\text{int}}$ to Joint Sigmoidal CBMs, Joint Logit CBMs, PCBMs, Residual PCBMs, large Hybrid CBMs ($k' = 500$), and CEMs trained without RandInt on the incomplete CUB task ($k = 22$).

Figure 8 shows that introducing $\mathcal{L}_{\text{int}}$ consistently improves intervenability across all models, often without degrading task or concept fidelity. These results support the claim that proper conditioning via sufficiency optimization can recover intervenability even in highly leaky architectures.

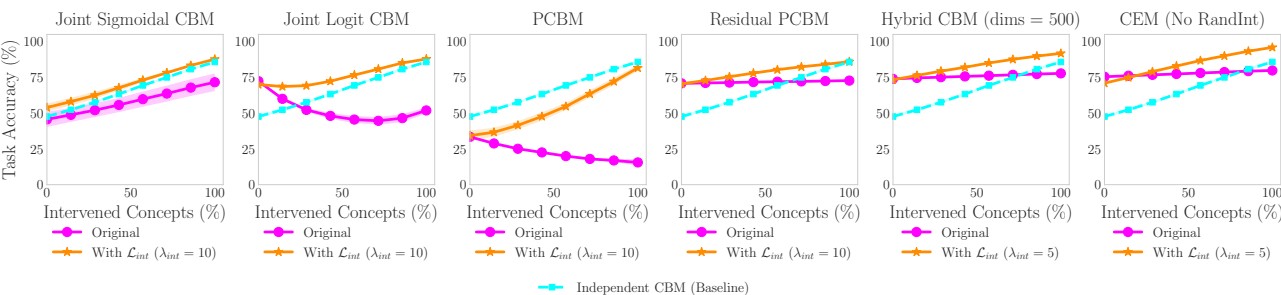

*Figure 8.* Extended results showing the effect of applying the sufficiency regularizer $\mathcal{L}_{\text{int}}$ to previously unintervenable leakage-enabling models on an *incomplete* version of CUB with $k = 22$ concepts.

## E. Verifying Trends on Additional Datasets

To further verify the generality of the claims in Section 6, we replicate our key intervention experiments on four additional datasets that span a range of incompleteness and noise regimes:

- **Full CUB** (complete but noisy concept set). We use the full class-standardized CUB (Wah et al., 2011) attribute set with $k = 312$ concepts. Several of these annotations are noisy, providing a useful stress test of our claims.

- **CIFAR10** (label-free concepts). Following Oikarinen et al. (2023), we extract $k = 143$ concepts for CIFAR10 via a label-free pipeline, processed and generated as in (Espinosa Zarlenga et al., 2025). Because these labels are obtained through foundation models, they are incomplete and noisy.

- **Incomplete Version of AwA2** (highly incomplete concept set). As in (Espinosa Zarlenga et al., 2025), we construct a highly incomplete version of Animals with Attributes 2 (AwA2) (Xian et al., 2018) by randomly selecting 9 out of the 85 available concept annotations.

- **SUN Attributes** (many-way, few-shot dataset). Inspired by prior works (Panousis et al., 2024; Vemuri et al., 2026), we explore the SUN Attributes (Patterson & Hays, 2012) dataset (717 classes), using the $k = 102$ class-level attribute annotations as concepts. This is a particularly difficult task for CMs given that its class-level concept annotations are noisy and contain inherently ambiguous concepts (e.g., "`mostly horizontal components`", "`moist`"), and there are only $\approx 20$ samples per class. These properties make SUN particularly prone to memorization.

For all of these tasks, we use the same models, dataset configurations, and hyperparameters as in (Espinosa Zarlenga et al., 2025). Therefore, here we focus on fine-tuning $\lambda_{\text{int}}$ for all baselines using the grid search described in App. B.

The only exception for how we select and train models is SUN. There, we use the frozen representations of a ResNet-101 trained on SUN as our backbone, and attach a learnable linear head to map them to the bottleneck. When training all models here, we (1) use a concept loss weight of $\lambda_{\text{concept}} = 10$ for all jointly trained CMs, (2) use a batch size of 256, (3) train all models using an Adam optimizer with learning rate $10^{-3}$ and early stopping with patience 20 epochs (for a maximum of 150 epochs), (4) use $k' = 100$ for the extra capacity in the bottleneck of the DNN and Hybrid CBM baselines (given the high number of output classes), and (5) select all other hyperparameters for all baselines by running the same model selection pipeline as in (Espinosa Zarlenga et al., 2025) for CUB tasks. To prevent residual channels in leaky CMs from memorizing training samples in SUN and to push the label predictor to rely on the concept-aligned components, we apply a high dropout rate (75%) to residual neurons during training, mirroring the residual-dropout strategy of Espinosa Zarlenga et al. (2025). We then report the effects of applying $\mathcal{L}_{\text{int}}$ to Black Box models and Hybrid CBMs, both with and without this dropout.

**Findings** As in Section 6, we observe that, across all additional tasks (Fig. 9), adding $\mathcal{L}_{\text{int}}$ makes leaky CMs that preserve leakage after being intervened on (e.g., Hybrid CBMs, CEMs) highly intervenable. In the most extreme incompleteness regime (AwA2), even models whose interventions destroy leakage (e.g., joint sigmoidal CBMs) surpass independent CBMs in unintervened task accuracy when trained with $\mathcal{L}_{\text{int}}$, although their initial accuracy drops as interventions are applied.

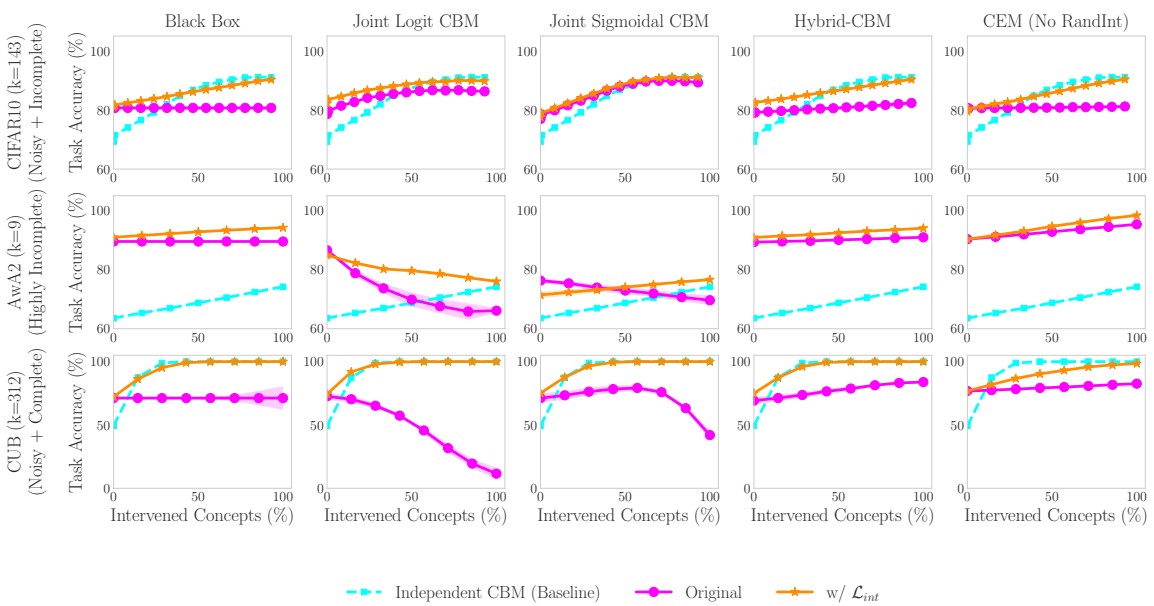

*(a)* Intervention curves on CIFAR10 (label-free concepts), an incomplete AwA2, and the full 312-concept CUB task.

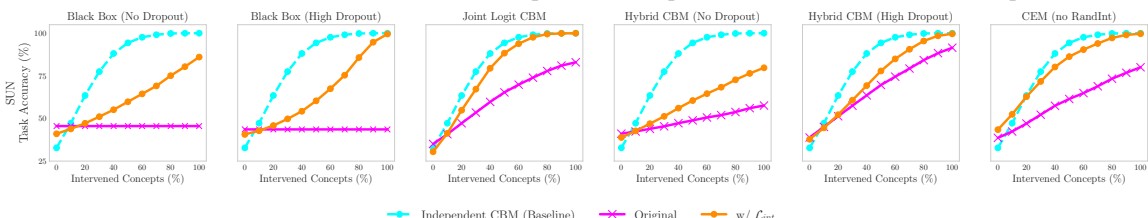

*(b)* Intervention curves on SUN Attributes (102 concepts, 717 classes). When applicable, we show curves for models trained with and without dropout on their residual components.

*Figure 9.* Intervention curves on additional tasks, verifying the results discussed in Section 6 for the incomplete version of CUB.

# F. Top-$m$ Weight Overlap

In Section 6.2 we compare the overlap in top-$m$ bottleneck-to-class weights between a well-conditioned Hybrid CBM $M_H$ and a (non-leaky) Independent CBM $M_I$. In that discussion, we specifically use $m = 5$ when presenting our arguments. Therefore, a natural concern is that the observed overlap is an artifact of the specific choice $m = 5$.

To rule this out, we sweep $m$ over a wide range and recompute the overlap $|\mathcal{T}_y(M_H) \cap \mathcal{T}_y(M_I)|$. As a reference we also report (i) the overlap between two Independent CBMs trained with different random seeds, which upper-bounds what we should expect from any two "interpretable-like" label predictors, and (ii) the overlap that an ill-conditioned Hybrid CBM ($\lambda_{\text{int}} = 0$) attains against $M_I$.

Our results, summarized in Figure 10, show that, as we vary $m$, the overlap in weight ranks between $M_H$ and $M_I$, shown by the orange line, and the overlap between two differently-seeded Independent CBMs, shown by the green line, remains close across the full range of $m$ for which the test remains discriminative (i.e., up until a random set of weights is indistinguishable from the overlap between two Independent CBMs with different seeds, shown in the dashed gray line). In contrast, the gap between the overlap of differently-seeded Independent CBMs and that of the ill-conditioned Hybrid CBM with $M_I$, shown by the blue line, is significantly different for most values of $m$. This suggests that the leaky but well-conditioned $M_H$ model allocates importance across concepts in a way that is statistically indistinguishable from the non-leaky $M_I$ model.

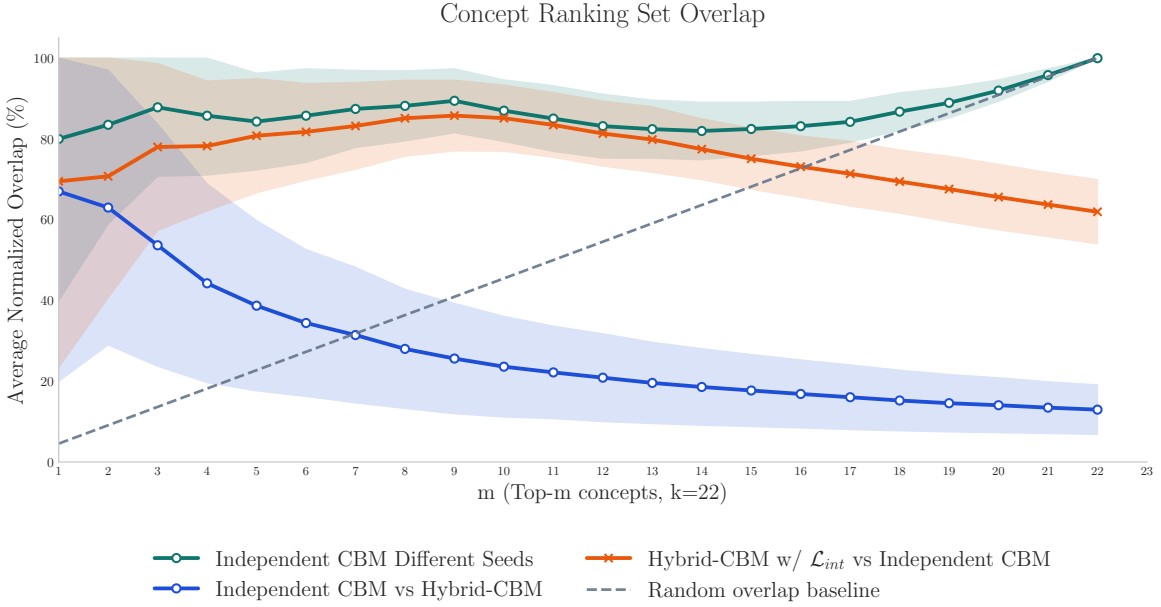

*Figure 10.* Top-$m$ weight overlap for (1) a properly-conditioned Hybrid CBM $M_H$ and a non-leaky Independent CBM $M_I$ (orange); (2) two Independent CBMs trained with different seeds (green); (3) an ill-conditioned Hybrid CBM and a non-leaky Independent CBM $M_I$ (blue); and (4) a random bottleneck subset and Independent CBM $M_I$ (gray dashed line, random baseline).

