# OpenReview forum: "Position: In Defense of Information Leakage in Concept-based Models"
_ICML.cc/2026/Position_Paper_Track — ICML 2026 Position Paper Track regular_

### Official Review · Reviewer_35zo · 2026-02-27

**Significance:** 2
**Argument Clarity:** 2
**Rating:** 4
**Confidence:** 4

**Questions:**

If interpretability claims against leakage are considered ill-posed due to a lack of measurement standards, what justifies treating intervenability and the chosen proxy metrics as sufficient evidence that leakage is benign?

How would the results change if the full set of 312 CUB attributes were used, or if the evaluation were extended to additional datasets? Does the argument for the necessity of leakage depend strongly on the degree of engineered incompleteness?

CBMs are often used for scientific discovery or auditing where "why" is more important than "what." Does the framework for "benign leakage" essentially convert CBMs back into black-box models with a conceptual "label" on top for the sake of task fidelity?

How do you distinguish between a model that is truly "intervenable" and one that is simply utilizing "leaked" correlations to maintain high accuracy regardless of the concept correction?

Can the sufficiency and localization conditions underlying benign leakage be measured or approximated directly in practice, beyond indirect proxy metrics?

**Alternative Views Section:**

Yes

**Compliance With Llm Reviewing Policy A Conservative:**

Affirmed.

**Discussion Potential:**

2

**Final Justification:**

I thank the authors for the thorough responses and the additional points provided during the discussion. The additional results further reinforce the position of the paper, along with the discussion on inference value and proxy metrics. I have adjusted my perspective accordingly and believe the paper is now in a much stronger state for publication.

**Paper Summary:**

This position papers argues in favor of allowing information leakage in Concept Models (CMs). The main premise is that eliminating information leakage is impractical for real-world settings, since "concept incompleteness" is common. In such cases, strictly enforcing concept purity may reduce the practical usefulness of CMs. To address this, the authors introduce the notion of "benign leakage", suggesting that allowing some concept-irrelevant information to leak into the prediction layer is necessary for achieving high task fidelity while maintaining intervenability.

The papers first present the prevailing alternative views to their position, then reframes the discussion by proposing a revised optimization objective view that emphasizes intervenability.  To support their position, the authors consider various CBM architectures which they train using a modified version of the CUB dataset with 112 concepts (instead of the full 312) to simulate a scenario of incomplete concept annotation. Their findings suggest that enforcing strict independence from leakage can reduce both predictive performance and the effectiveness of test-time interventions.

**Position:**

Yes

**Position In Title:**

Yes

**Related Work:**

2

**Strengths And Weaknesses:**

The paper focuses on a known tension in CBMs, i.e., the trade-off between strict "intepretability" in the form of concept purity and high task performance, and attempts to formalize information leakage as a feature rather than a bug. In this context, the authors place great emphasis on the interevenability aspect of CBM architectures, arguing that models should remain responsive to human correction (high level hints) even when utilizing auxiliary information; this is a relevant practical consideration, especially in cases where high quality annotated data are deemed difficult to obtain. The paper engages directly with a growing consensus in the literature that leakage is undesirable and should be minimized. Questioning this assumption is both intellectually valuable and practically relevant.

I find that there exists a contradictory logic when making claims with respect to intepretability. The paper initially argues that claims that leakage harms interpretability are "ill-posed" due to a lack of agreed-upon measurement lenses. However, it then evaluates its own position using a specific set of proxy metrics in a very controlled setting: measuring the ROC-AUC of concept prediction, comparing the top-5 weights, task fidelity and area under the intervention curve. It remains somewhat uncleared why these metrics are sufficient to declare leakage "benign" if interpretability itself is considered difficult to measure.

The score of empirical evaluation is very limited. The experiments are conducted exclusively on CUB, using a reduced subset of 112 concepts from the original 312 attributes. Although this setup may mirror some prior work, it provides a controlled way to simulate incompleteness, while raising questions about generality. It would strengthen the arguments that this position paper is making to explore whether the necessity of leakage persists with the full attribute set of across additional datasets, including multimodal or more complex domains.

The authors claim that the main purpose of statistical models is prediction; thus, the paper prioritizes almost exclusively predictive accuracy and intervention responsiveness, ignoring the other pillar of statistical models:  inference value. If a model uses leakage to make a prediction (even a benign one), a human is no longer able to distinguish why the model made that decision, which defeats the core purpose of using a Concept-based model in the first place.

While intervention curves are used as evidence that leakage does not undermine responsiveness, it remains somewhat unclear how to distinguish between a model that is genuinely reasoning through concept-aligned variables and one that maintains high accuracy because residual channels already encode the answer.

Comparing top-5 weights between leaky and non-leaky models is an interesting exploratory step, but the choice of threshold (e.g., top-5 rather than top-10, top-50 or a distributional comparison) appears somewhat arbitrary. A more systematic analysis of feature importance alignment might strengthen the interpretability argument.

**Support:**

2

---

> ### Author Rebuttal · Authors · 2026-03-29
>
> Thank you for the very valuable and constructive feedback! We are happy you found our position "intellectually valuable and practically relevant". Below, we address your concerns. Please do not hesitate to let us know if any of your concerns remain unsolved after this rebuttal.
>
>  ## (Potential misunderstanding) Use of CUB as an argumentation aid
>
> As this is a position paper, our experiments serve as argumentation aids, akin to prior position papers (e.g., [1, 2]). That said, we want to clarify a potential confusion: the 112-concept CUB is the **standard version of CUB in the CM literature** (e.g., [3–5] and most CM works). As described in App. B.1, *the concepts in this version are complete by construction*, each class having a unique concept profile. Hence, we simulate incompleteness by randomly subsampling $k=22$ concepts. We will make this point more explicit in Sec. 6.1.
>
> ## (Q1) Evaluating interpretability claims
>
> Please refer to our reply to W1 of reviewer gdx8.
>
> ## (W2) Declaring leakage "benign"
>
> We clarify that we call the form of leakage arising from localization and sufficiency "benign" not because it enables "interpretability" (as this is a difficult notion to measure), but because it still enables CMs to remain intervenable and accurate both in their task and in their concepts.
>
> ## (Q2) Generalization
>
> As discussed in our reply to Q1 of reviewer 7g9C, we observe consistent trends across (1) a noisy/unsupervised task (CIFAR10), (2) a highly incomplete dataset (in AwA2), and (3) a complete 312-concept CUB task. We will include these results in a new appendix.
>
> ## (Q3.1) Does benign leakage cause CBMs to lose inference value?
>
> We agree that inference value is an important property, but there is no reason why benign leakage should eliminate it. Three points worth noting:
>
> 1. In complete settings, $L_\text{int}$ incentivizes the CM to behave like a standard non-leaky CBM (as sufficiency is attainable without leakage). Hence, no inference value is lost here.
>
> 2. In incomplete settings, non-leaky CBMs may naturally lose their inference value: the model may not capture meaningful concept-to-task relationships, so non-leaky CBMs are not necessarily "useful" for inference in those instances either.
>
> 3. We respectfully disagree with the premise that the only purpose of CMs is to **fully** explain every prediction. Under that strict criterion, CMs would be disqualified from incomplete settings (e.g., most real-world tasks). More importantly, concept interventions themselves can retain considerable value if leakage is benign: they enable (1) high-level prior knowledge to be communicated (e.g., "There’s a bone spur on the X-ray") and (2) counterfactuals, both of which are not possible with feature manipulations in standard DNNs. We make this point in lines 90–108, col 2.
>
> We thank the reviewer for raising this alternative view of our position and will include it in Sec. 4.
>
> ## (Q3.2) Does benign leakage convert CBMs into black-box models with concepts on top?
>
> No. If that were the case, interventions would have no effect on task predictions (e.g., as in ill-conditioned Hybrid-CBMs in Figure 4, right). In contrast, well-conditioned leaky CMs (e.g., Hybrid-CBMs w/ $L_\text{int}$) show clear improvement in task accuracy as more concepts are intervened on (Fig 6). Responsiveness to interventions is the key distinguishing property.
>
> ## (Q4) Identifying "truly intervenable" models
>
> This is related to Q3. To summarise our answer, if a CM exploits leaked correlations to maintain accuracy “**regardless of concept corrections**”, this is detectable via intervention curves: task predictions would show no change or inconsistent changes under interventions (a “genuinely intervenable” CM would show improvements).
>
> ## (Q5) Measuring benign leakage and localization
>
> Please see our reply to Q2 of reviewer 7g9C.
>
> ## (W5) Use of top-5 weights
>
> Thank you for this suggestion. We used the top-5 concepts as it is a manageable number for an expert to review without mental overload. Nevertheless, as seen **[here](https://anonymous.4open.science/r/reb42/2.pdf)**, when we vary the number of top-m most-activating neurons considered, the overlap between the regularised Hybrid-CBM and the non-leaky ind CBM is very high, even for large $m$. In fact, it is similar to the overlap between two ind CBMs trained with different seeds up to the point where $m$ is high enough that this is no longer a useful test (i.e., compared to a random baseline). Hence, our finding is not specific to $m = 5$, and we will update our analysis by including this figure in Sec. 6.2.
>
> —
>
>
>  ## References
>
> 1. Hewitt et al. "We Can't Understand AI Using our Existing Vocabulary." ICML (2025).
> 2. Wilson. "Deep learning is not so mysterious or different." ICML (2025).
> 3. Koh et al. "Concept bottleneck models." ICML (2020).
> 4. Espinosa Zarlenga et al. "Concept embedding models." NeurIPS (2022).
> 5. Yuksekgonul et al. "Post-hoc concept bottleneck models." ICLR (2023).

---

> > ### Author Rebuttal · Reviewer_35zo · 2026-04-03
> >
> > The authors have provided a thorough response that addresses several of my initial concerns. Specifically, the additional analysis provided in response to other reviewers (e.g., experiments on CUB with 312 attributes, AwA2 and CIFAR-10) have largely resolved my questions regarding the generality of their findings. I would still be interested to know if the authors have considered the SUN dataset, as it is another common benchmark in this space, though I acknowledge the current additions already bolster the empirical argument significantly.
> >
> > Regarding the methodology, I maintain some reservations that the chosen proxy metrics (such as top-5 weights and intervention curves) may not be sufficient to fully distinguish "benign" leakage from more problematic correlations. However, for the purposes of a position paper, I find them acceptable as an initial assessment framework. I particularly appreciate the authors' discussion on inference value; this is a critical trade-off in concept-based modeling, and I strongly encourage the authors to incorporate these points into the final manuscript to provide a more balanced view.
> >
> > Since this is a position paper that offers an unconventional view capable of opening up a broader discussion in the field, along with the responses and the new results of the authors, I am leaning towards increasing my score. I will follow the discussion during this period and adjust accordingly.

---

> > > ### Author Response · Authors · 2026-04-05
> > >
> > > Thank you so much for going over our reply and for the encouraging update that you are leaning towards increasing your score. We are glad to read that our responses and new experiments resolved several of your concerns and that you believe this paper has the potential to open a broader discussion in the field. Below, we reply to the questions/points you raised above:
> > >
> > > ## Additional results on SUN
> > >
> > > We are pleased that our new results on three tasks have largely resolved your question about the generality of our results. From our perspective, these new results are best framed as further evidence that leakage does not imply that CMs are unintervenable, a common but mistaken claim in the literature. As such, we see these experiments, as well as the original CUB experiments in our manuscript, as aids for building a case in favor of (controlled) leakage.
> > >
> > > To further support our position, we followed your suggestion and ran experiments on SUN. Specifically, we used the 717-class SUN Attributes dataset [1], with the 102 class-level labels as concepts (as in [2, 3]). Our intervention results on SUN are summarized in **[this figure](https://anonymous.4open.science/r/reb42/3.pdf)**.
> > >
> > > To understand what these results reveal, it is worth noting that SUN is uniquely challenging for CMs to learn. This is because its class-level concept annotations are highly noisy and difficult to predict reliably from images (e.g., it includes difficult, ambiguous concepts such as “mostly horizontal components” and “moist”). Hence, we observed that (1) it is very difficult to train a vanilla concept predictor from a ResNet-101 backbone that attains a mean concept ROC-AUC above 85%, and (2) Independent CBMs trained on SUN perform very poorly on the task (i.e., 32.78%) while their label predictors can perfectly predict classes from the ground-truth concepts (as seen when the independent CBM is fully intervened on). Furthermore, the dataset has very few samples per class (~20 per class). All of these properties imply that it is difficult to train highly accurate CMs in SUN, as the model will be prone to memorization/overfitting when using the vanilla class-level concept labels.
> > >
> > > Because of this, we found that adding a heavy dropout rate on the residual activations of leaky models helped to prevent residual channels from memorizing training samples and to encourage the model to rely more on the concept representations (this is similar to the random dropout of residuals used in [4]). Therefore, in our results, we show the effects of applying $L_\text{int}$ to Black Box models and Hybrid-CBMs trained with and without a heavy dropout on their residual neurons (75% chance of dropping a residual neuron).
> > >
> > > Our results on SUN provide further evidence of what we argued in our position: when leakage is controlled (e.g., via a regularizer that encourages benign leakage, such as $L_\text{int}$), leaky models can become intervenable (while still achieving higher unintervened task accuracy than non-leaky models such as Independent CBMs). In SUN, we particularly see this with CEMs, where the introduction of $L_\text{int}$ made the model almost as intervenable as non-leaky independent CBMs, while attaining a significantly higher unintervened task accuracy than those models (i.e., 43.44% vs 32.78%). Note that this accuracy matches that of the Black Box ResNet-based baseline in [2].
> > >
> > > We thank the reviewer for suggesting exploring this task, and will include the SUN results in our manuscript’s appendix that discusses further evidence for the generality of the observations we use to support our position.
> > >
> > > ## Discussion on Inference Value
> > >
> > > We agree with the importance of this point! We commit to including a summary of this alternative view of our position in Section 4.
> > >
> > >
> > > ---
> > >
> > > Although we won’t be able to provide any further replies after this one, we hope this reply and the previous rebuttal have addressed most of your concerns. If that’s the case, we would sincerely appreciate it if your updated score could reflect this.
> > >
> > > Thank you for taking the time to review our paper, for participating in the review discussion, and for all the valuable feedback. We really appreciate it.
> > >
> > > ---
> > >
> > > ### References:
> > > 1. Patterson et al. "SUN attribute database: Discovering, annotating, and recognizing scene attributes." CVPR (2012).
> > > 2. Panousis et al. "Coarse-to-fine concept bottleneck models." NeurIPS  (2024).
> > > 3. Vemuri et al. "LogicCBMs: Logic-Enhanced Concept-Based Learning." WACV (2026).
> > > 4. Espinosa Zarlenga et al. "Avoiding leakage poisoning: Concept interventions under distribution shifts." ICML (2025).

---

### Official Review · Reviewer_gdx8 · 2026-03-09

**Significance:** 4
**Argument Clarity:** 4
**Rating:** 6
**Confidence:** 4

**Questions:**

Is L_int the same objective as training independent CBMs? Is the proposed modification to train with this loss together with another loss i.e. joint/sequential cbm loss? This should be described more clearly.

**Alternative Views Section:**

Yes

**Compliance With Llm Reviewing Policy A Conservative:**

Affirmed.

**Discussion Potential:**

3

**Final Justification:**

Maintaining my rating of 6, I believe this is a strong paper with very interesting contributions. After reading the rebuttal and other reviews, I do not have any significant remaining concerns.

**Paper Summary:**

This paper presents a view in defense of information leakage in concept-based models. In particular, the authors argue that not all information leakage is bad, but some of it is "benign" and can improve task performance without hurting interpretability. Based on this analysis, the authors also propose a simple intervention to improve interpretability (in particular intervene-ability) of concept-based model with information leakage.

**Position:**

Yes

**Position In Title:**

Yes

**Related Work:**

3

**Strengths And Weaknesses:**

Strengths:
- The main position is clearly defined and interesting as it goes against conventional stance in the field.
- The position is well justified, with both theoretical and experimental evidence supporting it
- The authors propose a simple training modification that helps promote benign leakage over harmful leakage
- I think this paper brings new insights into the field and will improve discussion and methods around concept leakage in the future
- Clearly written throughout

Weaknesses:
- The section against interpretability / safety issues with concept leakage is a bit weak, though these issues are hard to systematically address as they lack clear definitions as discussed by authors. Experimental results mostly focus on intervene-ability.

**Support:**

4

---

> ### Author Rebuttal · Authors · 2026-03-29
>
> Thank you for taking the time to review our work! We really appreciate the encouraging feedback. Specifically, we are glad you found our position "clearly defined", "interesting", and "well justified", and that you believe it goes "against conventional stance in the field". Below, we reply to your question and stated weakness. Please do not hesitate to let us know if you have any further concerns.
>
>  ## (Q1) Connection between $L_\text{int}$ and independent CBM training
>
> This is precisely the right intuition. One can think of $L_\text{int}$ regularization as encouraging "independent-like" behavior in a leaky CBM. In fact, as $\lambda_\text{int} \to \infty$, particularly for scalar CBMs, this essentially converges to independent training (the intuition extends to embedding-based representations, though the correspondence is less exact as leakage is preserved after these models' bottlenecks are intervened on).
>
> As you note, the proposed modification is to add a **heavy** $L_\text{int}$ penalty when training CMs to encourage benign leakage. The emphasis on "heavy" matters because, as argued in Sec. 6.1, we believe a strong penalty is needed to trigger benign leakage via the simplicity bias of DNN-based encoders.
>
> To make this clearer, we will explicitly draw the connection between our regularizer and independent training in Sec. 6.1, and will reiterate this point in a new discussion subsection in Sec. 7 (as suggested by reviewer 7g9C, W4).
>
> ## (W1) Use of proxy metrics for interpretability
>
> We agree that we rely on proxy metrics such as intervenability and concept ROC-AUC when evaluating prior “interpretability claims” that used similar (or exactly the same) metrics to claim that a model is “more interpretable” than another. However, we emphasize that we use intervention curves and these metrics in a subtly different way from prior work. Importantly, **we do not use these metrics to claim one model is more or less interpretable than another.** Rather, we use them to check whether the metrics that prior works cite as evidence **against** leaky models actually differ between models deemed "uninterpretable" (e.g., leaky Hybrid-CBMs) and those deemed "interpretable" (e.g., independent CBMs).
>
> Our experiments show these metrics do not necessarily differ across leaky and non-leaky methods, which is the basis of our claim that those interpretability arguments are ill-supported. For example, see lines 396-403, col 1, where we explicitly highlight that these same metrics used by previous works to evaluate “interpretability” do not appear to indicate a “preference” for a non-leaky model over a (well-conditioned) leaky model, as traditionally claimed in the literature (in fact, they appear to suggest the opposite conclusion if we follow their reasoning).
>
> To clarify this distinction, we will update Sec. 6.2 to explicitly state that we use proxy metrics for the purpose of validating previous claims that were made based on similar metrics (rather than making new "interpretability" claims ourselves).

---

> > ### Author Rebuttal · Reviewer_gdx8 · 2026-04-04
> >
> > Thank you for the response! After reading the rebuttal and other reviews, I do not have any significant remaining concerns and maintain my rating of strong accept.

---

> > > ### Author Response · Authors · 2026-04-07
> > >
> > > We would like to thank the reviewer for their encouraging feedback and review. We are glad that we addressed all significant concerns raised in their review. Thank you for all your time and for engaging in the rebuttal discussion!

---

### Official Review · Reviewer_dQSt · 2026-03-12

**Significance:** 2
**Argument Clarity:** 3
**Rating:** 4
**Confidence:** 4

**Questions:**

1. Is this position actually controversial? Leaky models like Hybrid CBMs and CEMs are already widely used in practice, which suggests many practitioners  already accept some degree of leakage. Have you encountered pushback on this view?

2. How do you expect benign leakage to behave in domains where concept incompleteness is much more severe than in CUB-200?

3. What made you choose the position track over the main track? The L_int regularizer and the experiments around it seem like they could stand on their own as a technical contribution.

**Alternative Views Section:**

Yes

**Compliance With Llm Reviewing Policy A Conservative:**

Affirmed.

**Discussion Potential:**

3

**Final Justification:**

The rebuttal has addressed my concerns. Therefore, I am raising my score to 4. However, I would not complain if this paper is rejected, given the uncertainty about how much interest the ICML community would have in this position, and because it seems to be a better fit for the main track rather  than the position paper track.

**Paper Summary:**

The paper challenges the common view that information leakage in concept-based models is always harmful. The authors argue that in realistic settings where concept annotations are incomplete, some leakage is actually necessary for building accurate and intervenable CMs. They formalize this as "benign leakage" (requiring sufficiency and localization), propose a regularizer L_int to encourage it, and run experiments on CUB-200 showing that leaky CMs can remain intervenable and interpretable.

**Position:**

Yes

**Position In Title:**

Yes

**Related Work:**

3

**Strengths And Weaknesses:**

**Strengths**

1. The position is clearly stated and goes against a widely held assumption in the CM literature. The paper does a good job walking through the existing arguments against leakage (intervenability, interpretability, safety) and pointing out where they fall short.

2. Benign leakage is cleanly defined. The sufficiency + localization decomposition makes intuitive sense and is backed by a formal proof (Theorem A.1).

**Weaknesses**

1. My main concern is that this feels more like a research paper than a position paper. Much of the paper's weight is on the L_int regularizer and the experiments validating it. The "position" part that the community should rethink leakage almost reads as motivation for a technical contribution. I wonder why this was not submitted to the main track instead.

2. Everything is on CUB-200. I understand it's a standard benchmark for CBMs, but a single dataset makes it hard to trust that the conclusions hold more broadly, especially in domains where concept incompleteness is far more severe (e.g., medical imaging, NLP).

3. The relevance to the broader ICML community feels limited. Concept-based models are a fairly specific subfield of XAI. The paper doesn't really discuss whether the idea of "benign leakage" has implications beyond CMs.

**Support:**

3

---

> ### Author Rebuttal · Authors · 2026-03-29
>
> Thank you for your insightful and constructive feedback! We are pleased you found our position "clearly stated" and that our argumentation did "a good job walking through the existing" counterarguments and "pointing out where they fall short". Below, we address your concerns. Please do not hesitate to let us know if any of your concerns remain unsolved after reviewing our replies.
>
>
> ## (Q1) Is this position controversial? Have you encountered pushback?
>
> Although determining whether our position is controversial requires some subjective judgment, we can say that the overwhelming consensus in the literature is that leakage is a negative phenomenon (e.g., see [1-4], as discussed in Sec. 4). This view does produce concrete pushback. For example, several CM papers contain statements such as:
>
> - "leakage makes soft CBMs unsuitable for tasks where interpretability or intervenability are required" [1]
> - "Information leakage … undermines interpretability" and "compromises the validity of interventions" [2]
> - "A concept-based model which displays leakage may therefore be unsafe for critical decision-making applications" [3]
> - "[leaky] bottlenecks may result in ineffective predictions that could prevent the use of CBMs in the wild" [4]
>
> This kind of pushback is also very common during reviewing/rebuttal (e.g., [A](https://openreview.net/forum?id=h61OIERd38), [B](https://openreview.net/forum?id=4FLes8q73Y), [C](https://openreview.net/forum?id=iSjqTQ5S1f)). Hence, we believe this position can lead to a healthy discussion of established beliefs.
>
> As you rightly note, CEMs and Hybrid-CBMs are common baselines. However, they are used as strong upper bounds on task performance, not because they are universally considered as interpretable as other CBM variants. Our position paper is intended to challenge the view that the inclusion of leaky baselines, when well-conditioned, should come with an “interpretability” caveat.
>
> ## (Q2 / W2) How does benign leakage behave in more incomplete domains?
>
> We observe that proper conditioning via a strong $ L_\text{int} $ term encourages benign leakage for, e.g., Hybrid CBMs, across multiple regimes. For concrete results, please see our response to Q1 of reviewer 7g9C, which shows consistent trends for Hybrid-CBMs with $L_\text{int}$ on CIFAR10, AwA2, and a 312-concept CUB.
>
> ## (Q3 / W1) Why is this better suited to the position track than the main track?
>
> We thank the reviewer for pointing out that this work may “live” somewhere in between the position track and the main track. Nevertheless, we believe it is better suited for the position track for two reasons.
>
> First, we believe the position itself, that leakage should not be avoided at all costs and can be controlled, is more important than the specific mechanism we introduce as a proof of concept. The position paper track is designed to encourage submissions of arguments that may not initially align with the current consensus (as we believe our position is), so we see this track as a better venue for reaching an audience receptive to arguments that may run counter to current narratives.
>
> Second, the mechanism we introduce ($L_\text{int}$) is a novel unifying view of several inductive biases in the literature (e.g., RandInt in CEMs, intervention-aware losses in IntCEMs, KL minimization in MCBMs, etc.). We therefore felt this regularizer was best framed as an initial proof-of-concept showing that it is indeed possible to allow for leakage without sacrificing elements people care about in CMs. For this, we drew inspiration from previous position papers that similarly introduce proofs of concept (e.g., [5]).
>
> ## (W3) Relevance to the broader ICML community
>
> CMs are arguably one of the fastest-moving fields in XAI, and seminal papers in this area (e.g., the CBM paper) have appeared at ICML. More importantly, the field overlaps heavily with representation learning, mechanistic interpretability (e.g., SAE representations), causality (e.g., interventions), and controllability (e.g., LLM steering and concept erasure). The presence of an entire ICLR26 workshop on "Unifying Concept Representation Learning" is further evidence of broad community interest.
>
> We agree, however, that we do not sufficiently emphasize the out-of-field implications. Therefore, we will update Sec. 7 to include a discussion of implications for adjacent areas, particularly for researchers building CMs on top of foundation models, LLMs, and multimodal systems, where some degree of leakage is practically unavoidable.
>
> —
>
> ## References
>
> 1. Havasi et al. "Addressing leakage in concept bottleneck models." NeurIPS (2022).
> 2. Almudévar et al. "There Was Never a Bottleneck in Concept Bottleneck Models" (2025).
> 3. Parisini et al. "Leakage and interpretability in concept-based models." arXiv (2025).
> 4. Sheth et al. "Auxiliary losses for learning generalizable concept-based models." NeurIPS (2023).
> 5. Hewitt et al. "We Can't Understand AI Using our Existing Vocabulary." ICML (2025).

---

> > ### Author Rebuttal · Reviewer_dQSt · 2026-04-03
> >
> > I thank the authors for the detailed rebuttal. The new experiments on address my concern about generality (the consistent trends across diverse incompleteness regimes are convincing). The cited evidence for the anti-leakage consensus in the literature also persuades me that this position is more controversial than I initially assumed.
> >
> > That said, two concerns remain partially open. First, while I accept the authors' framing of L_int as a proof-of-concept, the paper still reads more like a technical contribution with a position-paper wrapper, but this is a matter of presentation balance rather than substance, so I do not press it further.  Second, the promised broader discussion (connections to mechanistic interpretability, LLM steering, etc.) sounds valuable but is not yet in the manuscript, so I cannot evaluate it.

---

> > > ### Author Response · Authors · 2026-04-03
> > >
> > > Thank you for your reply! We are glad to read that we addressed Q1 and Q2. Below we comment on the two partially solved concerns outlined in your reply.
> > >
> > > Regarding the first concern, we will adjust our paper by framing our discussion of $\mathcal{L}_\text{int}$ as a unifying reinterpretation of existing inductive biases, so that it reads as part of the argument and as a concrete proof-of-concept of what we advocate.
> > >
> > > For the second concern, below we include the discussion subsection we will include in our final manuscript if accepted (potentially using the extra page).
> > >
> > > We thank you for your involvement in this review and hope this reply addresses your remaining partial concerns. If so, we would sincerely appreciate it if this could be reflected in your final updated review.
> > >
> > > ---
> > >
> > > # §7.1 - Leakage Outside of the Concept-based Literature
> > >
> > > Our paper's arguments have deep connections to adjacent areas of representation learning where leakage is equally unavoidable. Below, we discuss related research areas that have embraced leakage, thereby providing further evidence of the importance of **not dismissing leakage as something that can be removed**.
> > >
> > > **Sparse Autoencoders and Steering.** Sparse autoencoders (SAEs) [1, 2] are increasingly used to decompose latent representations of LLMs into human-interpretable features. A central goal of this line of work is *monosemanticity*, i.e., discovering features that respond to a single concept [2]. Our localization condition in benign leakage can be seen as a formal analogue of monosemanticity: information about concept $C_i$ that is predictive of $Y$ should be localized to a known component of the representation.
> > >
> > > Recognizing this connection has an important practical implication: *steering* on monosemantic SAE features is, in our framework, a form of benign leakage that has been successfully deployed in practice. Specifically, interventions on monosemantic features have been shown to produce targeted changes in LLM behavior [3, 4], which is precisely the intervenability guarantee that benign leakage encourages from localization. This suggests that the CM and the mechanistic interpretability literatures share the common goal of learning representations where concepts are localized. The difference, however, is in how these two fields approach this: the mechanistic interpretability literature treats leakage as an unavoidable property of powerful models and focuses instead on learning to localize it, rather than eliminate it as in the CM literature.
> > >
> > > We note that similar connections exist with works for *concept erasure* [5] and *representation engineering* [6, 7], where we are interested in modifying components of a transformer's internal representation such that we can control a specific concept (i.e., localize it) without damaging the model's downstream task utility (i.e., attain sufficiency). Hence, we hope that, by recognizing that leakage is not to be avoided at all costs, and instead focusing on *controlling* this leakage, future works in the CM literature may be more willing to explore methodologies where leakage is expected and required. Doing so would then enable CMs to be applied to the important setups where traditional steering and SAEs are used today.
> > >
> > > **Concept-based Foundation Models.** Label-free concept discovery pipelines [8-10] that build CMs on top of foundation models operate in precisely the regime where our argument for benign leakage applies most strongly: concept sets extracted from foundation models are inherently incomplete, and their representations encode far more information than that captured by the discovered concept set. Embracing benign leakage rather than attempting to eliminate it is, therefore, not only pragmatically necessary in this setting, but also the only path to scaling CMs. Therefore, we believe it is important to adjust the current narrative against leakage if the field is to enable the construction of powerful, concept-based foundation models. Otherwise, continuing to treat leakage as something that must be removed at all costs risks making this class of models unappealing to the community before their potential has been fully explored.
> > >
> > > ---
> > >
> > > ## References
> > >
> > > 1. Bricken et al. "Towards monosemanticity: ..." Anthropic Transformer Circuits Thread (2023).
> > > 2. Cunningham et al. "Sparse autoencoders find highly interpretable features in language models." ICLR (2024).
> > > 3. Templeton, et al. "Scaling monosemanticity: ..." Anthropic Transformer Circuits Thread (2024).
> > > 4. Arad et al. "Saes are good for steering–if you select the right features." EMNLP (2025).
> > > 5. Ravfogel et al. "Linear adversarial concept erasure." ICML (2022).
> > > 6. Zou et al. "Representation engineering: ..." arXiv (2023).
> > > 7. Li et al. "Inference-time intervention: ..." NeurIPS (2023).
> > > 8. Oikarinen et al. "Label-free concept bottleneck models." ICLR (2023).
> > > 9. Rao et al. "Discover-then-name: ..." ECCV (2024).
> > > 10. Yang et al. "Language in a bottle: ..." CVPR (2023).

---

### Official Review · Reviewer_7g9C · 2026-03-13

**Significance:** 3
**Argument Clarity:** 3
**Rating:** 4
**Confidence:** 3

**Questions:**

1.	How sensitive are the results to the choice of dataset and concept annotation quality?
2.	Can benign leakage be reliably identified or measured in practice for arbitrary architectures?
3.	How does the proposed regularization approach scale to large concept sets or high-dimensional representations?
4.	Are there scenarios where leakage could still meaningfully harm interpretability even when the proposed conditions are satisfied?

**Alternative Views Section:**

Yes

**Compliance With Llm Reviewing Policy A Conservative:**

Affirmed.

**Discussion Potential:**

3

**Final Justification:**

Thank you for the detailed and thoughtful rebuttal. The additional experiments across multiple datasets strengthen the empirical support, and the clarifications regarding the role of proxy metrics, scalability, and connections to prior methods improve the paper’s presentation and positioning.

While I still have some reservations, particularly regarding the reliance on proxy metrics and the lack of a fully principled definition of benign leakage in relation to interpretability, I appreciate the authors’ acknowledgment of these limitations and their framing as important directions for future work. Overall, I find the paper presents a clear, focused, and thought-provoking position on an important practical issue, and I believe it can stimulate useful discussion in the community.

**Paper Summary:**

This paper examines the role of information leakage in concept-based models, a class of interpretable machine learning models that ground their predictions in human-understandable concepts. The prevailing view in the literature is that leakage, when concept representations encode information beyond their intended semantics, is undesirable because it may compromise interpretability.  The authors challenge this perspective and argue that treating leakage as inherently harmful is often misguided. They claim that in realistic settings where concept annotations are incomplete, preventing leakage can significantly reduce model accuracy and practicality. To address this issue, the paper introduces the notion of benign leakage, which describes leakage that preserves both predictive performance and the ability to intervene on concepts.  The paper provides a theoretical characterization of benign leakage and proposes a training objective that encourages this behavior by minimizing task loss under full concept interventions. Experiments demonstrate that concept-based models trained with this objective can remain accurate even when leakage is present.

**Position:**

Yes

**Position In Title:**

Yes

**Related Work:**

2

**Strengths And Weaknesses:**

Strengths:
- The paper articulates a clear and focused position: information leakage in concept-based models is not necessarily harmful and can sometimes be beneficial. This thesis is consistently developed throughout the paper.
- The work addresses a common assumption in the concept-based modeling literature, that leakage should always be eliminated, and offers a systematic critique of that assumption. This reframing contributes to ongoing discussions about interpretability in machine learning.
- The introduction of benign leakage, along with formal conditions such as sufficiency and localization, provides a structured way to reason about leakage in concept representations. This formalization clarifies distinctions between harmful and potentially useful forms of leakage.
- The paper highlights an important practical issue: concept incompleteness in real-world datasets. By linking leakage to this problem, the authors provide a rationale for why strict leakage prevention may hinder the practical usefulness of concept-based models.
- The paper includes experimental results illustrating that models permitting controlled leakage can achieve strong task fidelity, concept fidelity, and intervenability. These results support the argument that leakage does not necessarily undermine interpretability metrics commonly used for concept-based models.

Weaknesses:
- The experimental analysis focuses primarily on a limited set of model architectures and datasets. While the experiments illustrate the main argument, broader evaluation across additional tasks and datasets would strengthen the generality of the claims.
- The paper critiques existing interpretability claims by noting the lack of a universal definition of interpretability. While this point is valid, the argument occasionally relies on proxy metrics without fully addressing whether those metrics capture the interpretability concerns raised in prior work.
- The proposed training objective encouraging benign leakage relies on intervention-based regularization. The paper could further discuss the computational cost and practical implications of applying this training procedure to large-scale models.
- Although the paper cites several studies on leakage and concept bottleneck models, a clearer comparison between the proposed framework and existing methods designed to mitigate leakage would help clarify how the proposed approach differs from or complements prior work.

**Support:**

3

---

> ### Author Rebuttal · Authors · 2026-03-29
>
> Thank you so much for your very valuable feedback! We are glad you found that our position "clear and focused" and that it highlights "an important practical issue". Below, we address your concerns. Please do not hesitate to let us know if you have any remaining concerns after reviewing our replies.
>
>
> ## (W2) Proxy metrics when discussing interpretability
>
> We use proxy metrics to verify the validity of previous claims about interpretability that are themselves supported by those proxy metrics. Hence, we do not use these metrics to claim a model is "more interpretable" than another. Rather, we use them to determine whether they differ between models deemed "uninterpretable" (e.g., Hybrid-CBMs) and those deemed "interpretable" (e.g., Ind. CBMs). For further details, please see our reply to W1 for gdx8.
>
> ## (W4) Relationship with other methods
>
> Lines 292–308, col 2 briefly discuss how minimizing $L_\text{int}$ relates to prior approaches. For clarity, we will briefly extend Sec. 7 to discuss how previous works use inductive biases that can be seen as implicitly minimizing $L_\text{int}$ (e.g., independent CBM training, RandInt in CEMs/ProbCBMs, intervention-aware loss in IntCEMs, the KL regularizer in MCBMs, the prior loss in MixCEMs, etc.).
>
>  ## (Q1) Sensitivity to dataset choice
>
> We used our incomplete CUB experiments as an **aid** for constructing our position (and we include experiments on a complete CUB in App. C). However, to show our argument generalizes, below we present new results on:
> - **CIFAR10**, where $k=143$ concepts were extracted using unsupervised pipelines [1] and interventions performed as in [2]. Because labels are obtained via LLMs and VLMs, they are incomplete and noisy.
> - A highly incomplete version of **AwA2** (randomly selecting $9/85$ concept labels).
> - A complete version of **CUB** using all original $k=312$ concepts (as suggested by reviewer 35zo). For this, we use the official class-standardized attribute annotations. We note that several concept labels here are noisy.
>
> Our results shown **[here](https://anonymous.4open.science/r/reb42/1.pdf)** show that adding $L_\text{int}$ makes leaky CMs that preserve leakage after interventions (e.g., Hybrid-CBMs) highly intervenable across all domains. Notice that these results also show that, in highly incomplete settings (AwA2), models that do not preserve leakage after an intervention (e.g., Joint CBMs), can outperform Ind. CBMs when trained with $L_\text{int}$, but their initial accuracy drops when intervened on because, unlike CEMs or Hybrid-CBMs, interventions destroy leakage (we emphasize this point in lines 294-297, col 2). All of these results will be included in a new appendix.
>
>  ## (Q2) Can benign leakage be identified?
>
> As a first-order approximation, the intervention curve AUC serves as a proxy for verifying localization: if a model does not localize concepts properly, interventions will have null or unpredictable effects. Similarly, comparing a CM's task accuracy to that of a black-box DNN trained on the same task enables one to approximate sufficiency.
>
> That said, without further assumptions about the CM's latent space or the concept-encoding distribution, it is difficult to propose a simple, general metric beyond this approach. We will therefore update our call to action to include a call for more principled ways to measure benign leakage.
>
> ## (Q3) Scalability
>
> Given $k$ $m$-dimensional concept representations, introducing $L_\text{int}$ requires one more pass of the (usually linear and small) task predictor on the intervened bottleneck. Assuming interventions have negligible costs, the computational cost from introducing $L_\text{int}$ scales as $\mathcal{O}(k m R(k, m))$, where $R$ is ratio of compute taken by task predictor $f(\cdot)$ vs the concept encoder $g(\cdot)$ for $k$ $m$-dimensional concept representations (usually $R(k, m) \ll 1$). For example, we observed that training a CEM on CUB with 312 concepts without $L_\text{int}$ took 76.23 ± 5.16 s/epoch, whereas with $ L_\text{int}$ it took 84.43 ± 1.00 s/epoch. This suggests that adding $L_\text{int}$ is a relatively inexpensive regularization term.
>
> As we believe this is an important discussion, we will update our manuscript to discuss the computational cost of introducing this regularizer when defining it in Section 5.1.
>
> ## (Q4) Can benign leakage damage interpretability?
>
> This is precisely the kind of questions we hope this position paper provokes. Our current answer is that we are not certain under which notions of interpretability one can claim that even benign leakage is undesirable. However, we include as part of our call to action (lines 433–435, col 2) an item asking future research to produce falsifiable criteria for when and under what definitions leakage is harmful to interpretability
>
>  ---
>
> ## References
>
> 1. Oikarinen et al. "Label-free concept bottleneck models." ICLR (2023).
> 2. Espinosa Zarlenga et al. "Avoiding leakage poisoning." ICML (2025).

---

> > ### Author Rebuttal · Reviewer_7g9C · 2026-04-03
> >
> > The authors provide thoughtful clarifications and additional experimental results across multiple datasets, which strengthen the empirical support. The discussion of proxy metrics, scalability, and connections to prior methods is also helpful.
> >
> > However, some of my concerns remain only partially resolved. In particular, the reliance on proxy metrics and the absence of a fully principled definition of benign leakage in relation to interpretability remain open questions. These issues concern the core framing of the paper. That said, I appreciate the authors’ acknowledgment of these limitations and their inclusion as part of the paper’s research agenda.

---

> > > ### Author Response · Authors · 2026-04-05
> > >
> > > Thank you so much for the follow-up and for taking the time to engage in the rebuttal process. We sincerely appreciate it. We are glad that our replies to Q1, Q3, Q4, and W4 were satisfactory. Below we provide further context on your remaining concerns about how we use proxy metrics.
> > >
> > > ## Why and how we use proxy “interpretability” metrics (W2 and Q2)
> > >
> > > We want to add more context on this concern, as there may be a subtle but important potential misreading of how we use proxy metrics. We emphasize that we use them specifically when debunking claims in the literature that rely on those same metrics to argue that leakage is harmful. This distinction needs to be more explicit in Section 6.2, and we will add a small preamble there to clarify it.
> > >
> > > As a concrete example, below we walk through one of the arguments we use to support our position that "**not all forms of leakage are malign**" [line 47, col 2]. The argument below, which summarizes the line of reasoning used in Section 6.1, showcases how and why we use certain proxy metrics.
> > >
> > > ---
> > >
> > > ### I. Re-stating prior intervenability claims
> > >
> > > Several works explicitly argue that leakage leads to unintervenable models (what we call "intervenability claims" in Section 4, lines 178–190, col 1). To quote directly from only *some* of the papers cited there:
> > >
> > > - "*Interpretability and intervenability are largely undermined in de facto technical solutions due to information leakage*” [1]
> > > - "*Information leakage … undermines interpretability" and "compromises the validity of interventions*" [2]
> > > - "*[leakage] results in corruptions in the concept predictions that impact the concept accuracy as well as our ability to intervene on the concepts*" [3]
> > > - "*A concept-based model which displays leakage may therefore be unsafe for critical decision-making applications*" [4]
> > >
> > > These claims are stated broadly, with the general implication that leakage must *always* render CMs unintervenable.
> > >
> > > ### II. Constructing a model with leakage
> > >
> > > In Section 6.1 (line 378, col 1) and Figure 6, we train a Hybrid CBM with $k' = 500$ unsupervised activations on an incomplete version of CUB using $L_\text{int}$. Because this model's task accuracy matches the black-box DNN's task accuracy on an *incomplete concept set*, we can conclude it must be exhibiting substantial leakage in its bottleneck.
> > >
> > > ### III. A counter-example to prior intervenability claims
> > >
> > > When we evaluate this model using **the same proxy metric that previous works used to argue that leakage is harmful**, namely intervention curves, we find that its task accuracy improves monotonically and outperforms an independently-trained sigmoidal CBM, a model widely considered to have minimal leakage. This directly contradicts the claims above, using those works' own standards.
> > >
> > > ### IV. The conclusion
> > >
> > > There must therefore exist forms of leakage that do not necessarily lead to unintervenable models (as we found a handful of counterexamples, shown in Figure 6). We call this "benign leakage" and posit it occurs when concept information is *localized* while representations are *sufficient* for the downstream task. We note that we *formalize* this form of leakage in Definition 5, and discuss how to encourage it *in practice* in Section 6.1. Crucially, however, the argument we make here does not claim our leaky model is "more interpretable" than a non-leaky one because it may have what we call "benign" leakage. Instead, it claims that the presence of leakage alone does not necessarily imply that a model must be unintervenable, as the previous consensus seemed to suggest.
> > >
> > > ---
> > >
> > > The same logic in the argument above applies to interpretability claims more broadly (Section 6.2). There, we show that the same leaky model achieves concept accuracies, intervention curves, and concept-to-task weight alignment that are statistically indistinguishable from the non-leaky independent CBM baseline. Hence, as we ask in lines 400–403, col 1, we want the community to ponder about under what perspective one can argue that a leaky CM is less interpretable than a non-leaky counterpart. Our position is that it is currently unclear whether such a perspective exists, given that, as our driving experiments show, task fidelity, concept fidelity, and intervenability (practical and desirable properties of CMs traditionally used to evaluate CMs) can all remain high even under high leakage.
> > >
> > > We hope this clarifies our use of proxy metrics. If so, we would sincerely appreciate it if this could be reflected in your final review.
> > >
> > > ---
> > >
> > > ## References
> > >
> > > 1. Sun et al. "Eliminating information leakage in hard concept bottleneck models with supervised, hierarchical concept learning." arXiv (2024).
> > > 2. Almudévar et al. "There Was Never a Bottleneck in Concept Bottleneck Models" ICLR (2026).
> > > 3. Havasi et al. "Addressing leakage in concept bottleneck models." NeurIPS (2022).
> > > 4. Parisini et al. "Leakage and interpretability in concept-based models." arXiv (2025).

---

### Decision · Program_Chairs · 2026-04-30

**Decision:**

Accept (regular)

**Comment:**

This paper was reviewed by four expert reviewers, receiving ratings of 6 (Strong Accept), and three 4s (Borderline Accept). All reviewers acknowledged the paper's clear and focused position challenging the conventional view that information leakage in concept-based models is inherently harmful. The introduction of "benign leakage" with formal conditions (sufficiency and localization) was recognized as a valuable contribution that reframes an important practical issue in the field.

Initial concerns centered on three main areas: (1) limited experimental scope, primarily using CUB-200, (2) reliance on proxy metrics for interpretability claims, (3) the balance between position argumentation and technical contribution (L_int regularizer). The authors provided a strong rebuttal with additional experiments on CIFAR10, AwA2, complete CUB with 312 concepts, and SUN datasets, demonstrating consistent trends across diverse regimes. These results strengthened the empirical support for their position.

Rev#7g9C and gdx8 appreciated the clarification that proxy metrics were used specifically to debunk prior claims made with those same metrics, rather than to assert new interpretability claims. Rev#dQSt noted the paper reads somewhat like a technical contribution with a position wrapper but accepted the framing of L_int as proof of concept. Rev#35zo's concerns about inference value and generality were largely addressed through additional experiments and discussion commitments. The rebuttal process was constructive, with reviewers acknowledging resolved concerns. Rev#gdx8 maintained their strong accept rating, while Rev#7g9C and dQSt confirmed partial resolution of concerns with appreciation for the authors' engagement. Rev#35zo indicated willingness to increase their score based on the comprehensive responses and new experimental results.

The AC recommends acceptance of this paper. It presents a thought-provoking position with potential to stimulate valuable discussion in the concept-based modeling community, backed by theoretical grounding and empirical validation across multiple datasets.